# Learning Dynamic Causal Graphs Under Parametric Uncertainty via Polynomial Chaos Expansions

**Liang Cao**
Department of Chemical and Biological Engineering
The University of British Columbia
Vancouver, BC V6T 1Z3, Canada
`clubc19@mail.ubc.ca`

## Abstract

Existing causal discovery methods are fundamentally limited by the assumption of a static causal graph, a constraint that fails in real-world systems where causal relationships dynamically vary with underlying system parameters. This discrepancy prevents the application of causal discovery in critical domains such as industrial process control, where understanding how causal effects change is essential. We address this gap by proposing a new paradigm that moves beyond static graphs to learn functional causal representations. We introduce a framework that models each causal link not as a static weight but as a function of measurable system parameters. By representing these functions using Polynomial Chaos Expansions (PCE), we develop a tractable method to learn the complete parametric causal structure from observational data. We provide theoretical proofs for the identifiability of these functional models and introduce a novel, provably convergent learning algorithm. On a large-scale chemical reactor dataset, our method learns the dynamic causal structure with a 90.9% F1-score, nearly doubling the performance of state-of-the-art baselines and providing an interpretable model of how causal mechanisms evolve.

## 1 Introduction

Industrial process control systems generate massive volumes of sensor data requiring automated analysis for optimization and predictive maintenance (Fang et al., 2024; Zhou et al., 2016). Understanding causal relationships between process variables is essential for root cause analysis, anomaly detection, and adaptive control strategies (Zhang et al., 2016). However, industrial processes exhibit unique challenges that violate assumptions of existing causal discovery methods: causal relationships may vary systematically with operating conditions, sensors exhibit complex multi-modal and heavy-tailed noise distributions, and safety-critical applications demand rigorous uncertainty quantification (Cao et al., 2025; Wang et al., 2025).

Many widely used causal discovery methods for observational data are formulated in terms of a single, static causal graph whose edge strengths do not depend on observed context or operating parameters, even though there is a growing body of work on time-varying and context-specific causal structures (Song et al., 2009; Huang et al., 2019). In reality, industrial causal effects are functions of measurable parameters. For instance, in chemical reactors, the influence of feed temperature on product quality depends strongly on catalyst activity, which degrades over time. Heat exchanger effectiveness varies with fouling levels, fundamentally altering thermal control loops. These parametric dependencies are not mere nuisances but contain critical information for process optimization and predictive maintenance. Our goal in this paper is therefore not to replace existing approaches to epistemic or aleatoric uncertainty, but to complement them with a representation in which each causal edge is an explicit function of a low-dimensional vector of operating parameters.

The field of causal discovery has evolved through three major directions, each with distinct limitations for industrial applications. Constraint-based methods such as the Peter-Clark (PC) algorithm

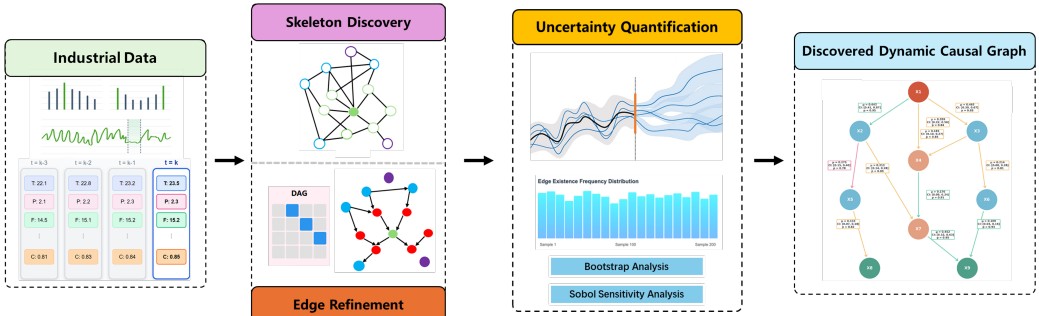

Figure 1: Overview of the polynomial chaos theory for causal discovery framework in dynamic uncertainty systems.

(Spirtes et al., 2000), Fast Causal Inference (FCI) (Spirtes et al., 1999), and Really Fast Causal Inference (RFCI) (Colombo et al., 2012) rely on conditional independence tests to infer causal structures. While theoretically sound, these methods struggle with finite sample sizes and become computationally intractable for high-dimensional industrial data with hundreds of sensors. Recent advances like PC-stable (Colombo & Maathuis, 2014) improve stability but fail when faced with complex noise distributions common in industrial sensors.

Score-based methods including Greedy Equivalence Search (GES) (Chickering, 2002) and Fast Greedy Equivalence Search (FGES) (Ramsey et al., 2017) optimize scoring functions over possible Directed Acyclic Graph (DAG) structures. The breakthrough NOTEARS algorithm (Zheng et al., 2018) reformulated structure learning as continuous optimization with differentiable acyclicity constraints, spawning variants like Directed Acyclic Graph - Graph Neural Network (DAG-GNN) (Yu et al., 2019) for nonlinear relationships, Reinforcement Learning - Bayesian Information Criterion (RL-BIC) (Zhu et al., 2020) using reinforcement learning. In their most common instantiations, these approaches return a single best-scoring DAG and point estimates of edge weights; uncertainty over graphs and parameters is typically handled by separate Bayesian or bootstrap procedures (e.g., Lorch et al., 2021; Cundy et al., 2021) rather than being integrated with an explicit model of how edge strengths vary with operating parameters. This limitation is particularly critical for safety-critical industrial applications where confidence in causal recommendations directly impacts operational decisions and safety outcomes.

Functional causal models exploit asymmetries in data distributions for identifiability. The Linear Non-Gaussian Acyclic Model (LiNGAM) (Shimizu et al., 2006) proved that linear models with non-Gaussian noise yield unique causal structures, later extended to DirectLiNGAM (Shimizu et al., 2011) and Vector Autoregressive LiNGAM (VAR-LiNGAM) (Hyvärinen et al., 2010) for time series. Nonlinear extensions include additive noise models (ANM) (Hoyer et al., 2008), post-nonlinear models (PNL) (Zhang & Hyvärinen, 2009), and the general identifiable functional causal model framework (Peters et al., 2014). However, in their standard form these models do not represent how causal effects change as an explicit function of observed operating parameters.

Recent industrial applications (Sui et al., 2025; Runge et al., 2019; Yang et al., 2025) have highlighted these limitations, often requiring extensive preprocessing or domain-specific modifications lacking theoretical justification. Bayesian approaches such as Differentiable Bayes for Structure Learning (DiBS) (Lorch et al., 2021) and Bayesian Causal Discovery with Neural Networks (BCD Nets) (Cundy et al., 2021) explicitly quantify posterior uncertainty over graphs and parameters, addressing epistemic uncertainty due to finite data, but they still treat each edge as static and do not model how its strength varies with operating parameters.

Polynomial Chaos Expansion (PCE), introduced by Wiener (Wiener, 1938) and generalized by Xiu (Xiu & Karniadakis, 2002), provides a mathematically rigorous framework for representing and propagating uncertainty through complex systems. PCE has been successfully applied in uncertainty quantification for engineering systems (Sudret, 2008), sensitivity analysis (Crestaux et al., 2009), and process monitoring (Cao et al., 2026). Recent algorithmic advances including sparse PCE (Jiang et al., 2025), adaptive basis selection (Dai et al., 2025), and multi-fidelity approaches (Liu et al., 2021) have made PCE computationally tractable for high-dimensional problems. Despite this

success in forward uncertainty propagation, PCE has, to the best of our knowledge, not yet been systematically exploited for causal discovery.

This paper introduces PCT-CD, bridging uncertainty quantification and causal discovery. Figure 1 provides an overview of the proposed framework. Our contributions are summarized as follows:

1. **From Static Graphs to Dynamic Functions:** We formalize an industrial structural causal model in which a single underlying DAG is equipped with edge weights that are explicit functions of operating conditions, and we prove identifiability of these parametric mechanisms under mild assumptions.

2. **An End-to-End Solution for Dynamic Systems:** We propose PCT-CD, an integrated algorithm specifically designed for parametric causal discovery. It translates complex process data into an interpretable model of how causal links evolve, providing actionable intelligence beyond simple correlation.

3. **Demonstrated Real-World Viability:** We empirically validate PCT-CD on controlled benchmark with parameter-varying mechanisms. PCT-CD achieves high F1-scores and equips engineers with uncertainty-aware tools that are essential for making robust decisions in high-stakes environments.

The remainder of this paper is organized as follows. Section 2 details the PCT-CD methodology, including parametric SEM formulation, PCE representation, and our novel conditional independence test. Section 3 establishes theoretical foundations with identifiability and convergence proofs. Section 4 validates our approach on synthetic benchmark with controlled parametric variation and an industrial process dataset, comparing against a broad set of baseline methods. Section 5 concludes with implications for industrial deployment and future research directions.

## 2 METHODOLOGY

Our proposed framework, PCT-CD, introduces a novel paradigm for causal discovery by explicitly modeling how causal relationships vary as functions of measurable system parameters. This is achieved by integrating the theory of PCE into a hybrid structure learning algorithm. While Bayesian and bootstrap-based methods typically quantify epistemic uncertainty arising from finite data (for example via posterior distributions over static graphs and parameters), PCT-CD is designed to address *parametric* uncertainty by representing causal edges as explicit functions of measurable system parameters, and is conceptually complementary to these existing approaches.

The methodology unfolds in four stages: first, we formulate a Structural Equation Model (SEM) where causal coefficients are functions of a parameter vector $\boldsymbol{\xi}$. Second, we represent these functions using PCE, transforming the non-parametric problem into a tractable parametric one. Third, we develop a novel conditional independence test tailored to this representation to discover an initial causal skeleton. Finally, we refine this structure and quantify edge strengths using a score-based optimization with a natural gradient approach, ensuring both accuracy and computational efficiency.

### 2.1 PROBLEM FORMULATION

We consider a complete probability space $(\Xi, \mathcal{F}, \mathbb{P})$ where all random quantities are defined. The core innovation of our framework is the explicit modeling of parametric uncertainty through a random vector $\boldsymbol{\xi} \in \Xi \subset \mathbb{R}^d$. This vector represents known, measurable operating conditions (e.g., ambient temperature, catalyst age, feedstock quality) with a joint probability distribution $\mu_\xi$ that has finite moments of all orders. In our theoretical analysis we assume that $\mu_\xi$ is known so that a standard PCE basis adapted to $\mu_\xi$ can be chosen; in practice, when only samples of $\boldsymbol{\xi}$ are available, an empirical orthogonal basis can be constructed from the observed parameter values (see Appendix for details). This formulation emphasizes a dimension that is often implicit in causal discovery, where operating conditions are typically treated as fixed and uncertainty is mainly modeled as arising from finite data and stochastic noise.

We observe $n$ process variables, collected in a vector $\mathbf{X} = (X_1, \ldots, X_n)^T \in \mathbb{R}^n$. We assume that these variables are generated by a linear SEM (equivalently, a linear structural causal model in the sense of structural causal inference) where the causal relationships are functions of the parameter

vector $\boldsymbol{\xi}$:

$$X_i = \sum_{j \in \mathbf{PA}_i} b_{ij}(\boldsymbol{\xi}) X_j + \epsilon_i, \quad i = 1, \ldots, n \tag{1}$$

where $\mathbf{PA}_i \subset \{1, \ldots, n\} \setminus \{i\}$ denotes the set of causal parents of variable $X_i$, the functions $b_{ij}(\boldsymbol{\xi}) \in L^2(\Xi)$ are unknown, square-integrable functions capturing the parameter-dependent causal effects, and $\epsilon_i$ are mutually independent, centered, sub-Gaussian noise terms. The underlying causal structure forms a DAG $\mathcal{G} = (V, E)$, where $V = \{1, \ldots, n\}$ and an edge $(j, i) \in E$ exists if and only if $j \in \mathbf{PA}_i$. We make the standard assumptions of causal sufficiency (no unmeasured common causes) and faithfulness (all conditional independencies in the data are consequences of d-separation in $\mathcal{G}$). Throughout, the edge set $E$ does not depend on $\boldsymbol{\xi}$; only the edge weights $b_{ij}(\boldsymbol{\xi})$ vary with operating conditions. We assume $m$ i.i.d. samples $\{(\mathbf{X}^{(t)}, \boldsymbol{\xi}^{(t)})\}_{t=1}^m$ from this model, where $t$ indexes samples rather than time.

## 2.2 POLYNOMIAL CHAOS REPRESENTATION

The central challenge is to learn the functions $b_{ij}(\boldsymbol{\xi})$, the true causal strength varying with system parameters $\boldsymbol{\xi}$. We address this by representing each causal coefficient function using a PCE. For many choices of $\mu_\xi$, including the classical Wiener–Askey scheme, there exists a corresponding basis of orthogonal multivariate polynomials $\{\Psi_\alpha(\boldsymbol{\xi})\}_{\alpha \in \mathbb{N}^d}$ adapted to $\mu_\xi$ (e.g., Xiu & Karniadakis, 2002; Sudret, 2008; Crestaux et al., 2009). Common examples include Hermite polynomials for Gaussian parameters, Legendre for uniform and Laguerre for exponential.

Any square-integrable function $b_{ij}(\boldsymbol{\xi})$ can be expanded in this basis. By truncating the expansion at a total polynomial degree $N_p$, we obtain a finite-dimensional approximation:

$$b_{ij}(\boldsymbol{\xi}) \approx \sum_{\alpha \in \mathcal{A}_{N_p}} \theta_{ij,\alpha} \Psi_\alpha(\boldsymbol{\xi}) \tag{2}$$

where $\mathcal{A}_{N_p} := \{\alpha \in \mathbb{N}^d : |\alpha| = \sum_{k=1}^d \alpha_k \leq N_p\}$ is the set of multi-indices, and the coefficients $\theta_{ij,\alpha}$ are the spectral projections of the function onto the basis, given by $\theta_{ij,\alpha} = \langle b_{ij}(\boldsymbol{\xi}), \Psi_\alpha(\boldsymbol{\xi}) \rangle_{L^2} / \langle \Psi_\alpha^2 \rangle_{L^2}$. The cardinality of the basis is $P = |\mathcal{A}_{N_p}| = \binom{N_p + d}{d}$. This representation converts the infinite-dimensional problem of learning functions $b_{ij}(\boldsymbol{\xi})$ into a finite-dimensional problem of estimating the spectral coefficients $\theta_{ij,\alpha}$.

For functions that are continuously differentiable $s$ times, the spectral error decays polynomially: $\|b_{ij} - \Pi_{N_p} b_{ij}\|_{L^2} \leq C N_p^{-s}$, where $\Pi_{N_p}$ is the projection operator. For analytic functions, which are common in physical systems, convergence is exponential: $\|b_{ij} - \Pi_{N_p} b_{ij}\|_{L^2} \leq C \exp(-\gamma N_p^{1/d})$ (e.g., Sudret, 2008; Crestaux et al., 2009). For high-dimensional parameter spaces ($d \gg 1$), the basis size $P$ can become computationally prohibitive. We employ hyperbolic truncation schemes, which prioritize low-order interaction terms and significantly reduce the basis size while often retaining high accuracy for functions with decaying importance of higher-order interactions (Jiang et al., 2025; Dai et al., 2025).

## 2.3 PCT-CONDITIONAL INDEPENDENCE TEST

Once we have established the PCE representation, our initial goal is to identify the causal skeleton. Standard conditional independence (CI) tests that operate on the marginal distribution of $(X_A, X_B, X_Z)$ can fail in the presence of parameter-varying mechanisms: a causal relationship may be strong for many values of $\boldsymbol{\xi}$ but cancel out in the marginal (e.g., sign-changing effects), leading marginal CI tests to falsely conclude independence. We therefore test conditional dependence *as a function of* the operating parameters.

Let

$$C_{AB|Z}(\boldsymbol{\xi}) := \mathrm{Cov}(X_A, X_B \mid X_Z, \boldsymbol{\xi})$$

denote the conditional covariance function over the parameter space.

**Definition 1** (PCT-conditional independence). *We say that $X_A$ and $X_B$ are PCT-conditionally independent given $X_Z$ if*

$$\|C_{AB|Z}\|_{L^2(\mu_\xi)}^2 := \mathbb{E}_{\boldsymbol{\xi}}[C_{AB|Z}(\boldsymbol{\xi})^2] = 0, \tag{3}$$

*equivalently, $C_{AB|Z}(\boldsymbol{\xi}) = 0$ for $\mu_\xi$-almost every $\boldsymbol{\xi}$.*

Under mild moment conditions, $C_{AB|Z}(\boldsymbol{\xi}) \in L^2(\mu_\xi)$ admits a (truncated) polynomial chaos expansion

$$C_{AB|Z}(\boldsymbol{\xi}) \approx \sum_{\alpha \in \mathcal{A}_{N_p}} C_{AB|Z,\alpha} \Psi_\alpha(\boldsymbol{\xi}), \qquad P := |\mathcal{A}_{N_p}|. \tag{4}$$

By orthogonality of $\{\Psi_\alpha\}$, the null hypothesis is equivalent (under truncation) to the vector of PCE coefficients being zero:

$$H_0: \ \boldsymbol{C}_{AB|Z} := (C_{AB|Z,\alpha})_{\alpha \in \mathcal{A}_{N_p}} = \boldsymbol{0}.$$

To estimate $C_{AB|Z}(\boldsymbol{\xi})$, we first remove the dependence on $(X_Z, \boldsymbol{\xi})$ by regressing $X_A$ and $X_B$ on the *interaction features* $\{X_k \Psi_\alpha(\boldsymbol{\xi})\}_{k \in Z, \alpha \in \mathcal{A}_{N_p}}$:

$$X_A^{(t)} \approx \sum_{k \in Z} \sum_{\alpha \in \mathcal{A}_{N_p}} \beta_{A,k,\alpha} X_k^{(t)} \Psi_\alpha(\boldsymbol{\xi}^{(t)}) + r_{A|Z,\boldsymbol{\xi}}^{(t)}, \quad X_B^{(t)} \approx \sum_{k \in Z} \sum_{\alpha \in \mathcal{A}_{N_p}} \beta_{B,k,\alpha} X_k^{(t)} \Psi_\alpha(\boldsymbol{\xi}^{(t)}) + r_{B|Z,\boldsymbol{\xi}}^{(t)}.$$

The residual product $r_{A|Z,\boldsymbol{\xi}}^{(t)} r_{B|Z,\boldsymbol{\xi}}^{(t)}$ is then an empirical estimate of the conditional covariance signal at $\boldsymbol{\xi}^{(t)}$. Define the basis vector $\boldsymbol{\psi}(\boldsymbol{\xi}) := (\Psi_\alpha(\boldsymbol{\xi}))_{\alpha \in \mathcal{A}_{N_p}} \in \mathbb{R}^P$, and

$$\mathbf{v}^{(t)} := r_{A|Z,\boldsymbol{\xi}}^{(t)} r_{B|Z,\boldsymbol{\xi}}^{(t)} \boldsymbol{\psi}(\boldsymbol{\xi}^{(t)}) \in \mathbb{R}^P, \qquad \hat{\boldsymbol{C}} := \frac{1}{m} \sum_{t=1}^{m} \mathbf{v}^{(t)}.$$

Let $\hat{\Sigma}$ be the sample covariance of $\{\mathbf{v}^{(t)}\}_{t=1}^m$:

$$\hat{\Sigma} := \frac{1}{m-1} \sum_{t=1}^{m} (\mathbf{v}^{(t)} - \hat{\boldsymbol{C}})(\mathbf{v}^{(t)} - \hat{\boldsymbol{C}})^\top.$$

Under $H_0$ and standard regularity conditions (i.i.d. samples conditional on $\boldsymbol{\xi}$, finite fourth moments, and nonsingularity of $\Sigma$), a multivariate CLT yields $\sqrt{m} \hat{\boldsymbol{C}} \Rightarrow \mathcal{N}(\boldsymbol{0}, \Sigma)$. Consequently, the Wald statistic

$$T_{PCT} := m \hat{\boldsymbol{C}}^\top \hat{\Sigma}_{\text{reg}}^{-1} \hat{\boldsymbol{C}} \xrightarrow{d} \chi_{\text{df}}^2, \qquad \hat{\Sigma}_{\text{reg}} := \hat{\Sigma} + \varepsilon_{\text{reg}} \mathbf{I}, \tag{5}$$

where $\text{df} = \text{rank}(\hat{\Sigma})$ (typically $\text{df} = P$) and $\varepsilon_{\text{reg}} > 0$ is a small ridge term for numerical stability. We use this test as the CI oracle inside a PC-style skeleton search (Algorithm 1).

## 2.4 SCORE-BASED LEARNING WITH NATURAL GRADIENT

Although constraint-based methods are effective for skeleton discovery, they can be unstable with finite data. We therefore use the output of the constraint-based phase as an initialization for a more robust score-based optimization. We formulate structure learning as the optimization of a penalized likelihood score over the space of DAGs and PCE coefficients and define the PCT-BIC score as:

$$\mathcal{S}(E, \Theta) = \frac{1}{2\sigma_\epsilon^2} \sum_{t=1}^{m} \sum_{i=1}^{n} \left( X_i^{(t)} - \sum_{j \in \mathbf{PA}_i} \sum_{\alpha \in \mathcal{A}_{N_p}} \theta_{ij,\alpha} \Psi_\alpha(\boldsymbol{\xi}^{(t)}) X_j^{(t)} \right)^2 + \lambda \|(E, \Theta)\|_0 \tag{6}$$

where $\Theta = \{\theta_{ij,\alpha}\}$ is the collection of all PCE coefficients. The group sparsity penalty $\|(E, \Theta)\|_0 = \sum_{i,j} \mathbf{1}\{\|\boldsymbol{\theta}_{ij}\|_2 > 0\}$ encourages sparse DAGs by penalizing the number of non-zero causal links. Here $\lambda > 0$ is a regularization weight and in practice we treat $\lambda$ as a tunable hyperparameter.

Optimizing this score is challenging due to the combinatorial nature of the graph space and the high dimensionality of the parameter space $\Theta$. We employ a greedy search strategy combined with efficient gradient-based optimization of the coefficients for a given graph structure, accepting edge additions or deletions only when they preserve acyclicity of $\mathcal{G}$.

**Natural-gradient / preconditioned updates for coefficient optimization.** For a fixed DAG $\mathcal{G}$ (hence fixed parent sets $\{\mathbf{PA}_i\}$), the sparsity penalty in equation 6 is constant, and optimizing over $\Theta$ reduces to $n$ separate least-squares problems, one per node $i$. For each node $i$, define the design matrix $\Phi_i \in \mathbb{R}^{m \times (|\mathbf{PA}_i|P)}$ with entries

$$[\Phi_i]_{t,(j,\alpha)} := X_j^{(t)} \Psi_\alpha(\boldsymbol{\xi}^{(t)}), \qquad j \in \mathbf{PA}_i, \ \alpha \in \mathcal{A}_{N_p},$$

and let $\boldsymbol{\theta}_i \in \mathbb{R}^{|\mathbf{PA}_i|P}$ stack $\{\theta_{ij,\alpha}\}_{j \in \mathbf{PA}_i, \alpha}$. Then the data-fit term for node $i$ is

$$\mathcal{L}_i(\boldsymbol{\theta}_i) = \frac{1}{2\sigma_\epsilon^2}\|\mathbf{x}_i - \Phi_i\boldsymbol{\theta}_i\|_2^2, \qquad \mathbf{x}_i = (X_i^{(1)}, \ldots, X_i^{(m)})^\top.$$

Under a linear-Gaussian (or quasi-likelihood) view, the empirical Fisher information is

$$\hat{\mathbf{F}}_i = \frac{1}{\sigma_\epsilon^2 m}\Phi_i^\top\Phi_i,$$

which is generally *not diagonal* because the regressors depend on $\boldsymbol{\xi}$ and can be correlated. We therefore apply a Fisher-preconditioned update (natural gradient)

$$\boldsymbol{\theta}_i \leftarrow \boldsymbol{\theta}_i - \eta\,(\Phi_i^\top\Phi_i + \varepsilon_{\mathrm{reg}}\mathbf{I})^{-1}\Phi_i^\top(\Phi_i\boldsymbol{\theta}_i - \mathbf{x}_i),$$

with a small ridge $\varepsilon_{\mathrm{reg}} > 0$ for numerical stability and warm starts as the graph is updated during greedy search.

The complete PCT-CD algorithm, summarized in Algorithm 2 in the appendix, integrates these components into a multi-phase procedure that ensures both structural accuracy and robust parameter estimation. This provides not only the final graph and functional relationships but also confidence intervals for causal strengths and probabilities for the existence of each edge.

## 3 THEORETICAL ANALYSIS

In this section, we establish the theoretical foundations of the PCT-CD framework. We prove that under reasonable conditions, the true parametric causal DAG is uniquely identifiable from observational data. Furthermore, we provide finite-sample guarantees for the recovery of the causal structure and analyze the convergence properties of our optimization procedure. Formal statements and proofs of the main results are deferred to the Appendix for clarity.

### 3.1 ASSUMPTIONS AND PRELIMINARIES

**Assumption 1** (Data-generating process). *The variables $\mathbf{X} = (X_1, \ldots, X_n)$ obey the SEM equation 1 with a DAG $\mathcal{G}$, where each $b_{ij}(\boldsymbol{\xi}) \in L^2(\mu_\xi)$ admits the chaos expansion equation 2 with truncation bias controlled by polynomial convergence theory. The parameter vector $\boldsymbol{\xi}$ is independent of noises $\{\epsilon_i\}$ and has a known distribution $\mu_\xi$ with finite moments of all orders.*

**Assumption 2** (Noise). *The disturbances $\epsilon_i$ are mutually independent, centered, sub-Gaussian with proxy $\sigma_\epsilon^2$ and finite fourth moments. Moreover, at most one $\epsilon_i$ is Gaussian (equivalently, all but at most one are non-Gaussian).*

**Assumption 3** (Faithfulness and stability). *For $\mu_\xi$-almost every $\boldsymbol{\xi}$, the conditional distribution $P(\mathbf{X} \mid \boldsymbol{\xi})$ is faithful to the DAG $\mathcal{G}$. The operator norm satisfies $\mathbb{E}[\|\mathbf{B}(\boldsymbol{\xi})\|_{\mathrm{op}}] < 1$, where $\mathbf{X} = \mathbf{B}(\boldsymbol{\xi})\mathbf{X} + \boldsymbol{\epsilon}$ is the matrix form with $[\mathbf{B}(\boldsymbol{\xi})]_{ij} = b_{ij}(\boldsymbol{\xi})$ for $j \in \mathbf{PA}_i$ and zero otherwise.*

**Assumption 4** (Design regularity). *Fix a truncation order $N_p$ and basis size $P = |\mathcal{A}_{N_p}|$, and let $s$ be an upper bound on in-degree. For each node $i$ and any candidate parent set $S_i$ with $|S_i| \leq s$, define the interaction feature vector*

$$\boldsymbol{\phi}_{i,S_i} := \big(X_j\,\Psi_\alpha(\boldsymbol{\xi})\big)_{j \in S_i,\,\alpha \in \mathcal{A}_{N_p}} \in \mathbb{R}^{|S_i|P},$$

*and the population Gram matrix $\mathbf{G}_{i,S_i} := \mathbb{E}[\boldsymbol{\phi}_{i,S_i}\boldsymbol{\phi}_{i,S_i}^\top]$. Assume there exists a constant $\gamma > 0$ such that*

$$\lambda_{\min}(\mathbf{G}_{i,S_i}) \geq \gamma \quad \text{for all } i \text{ and all } S_i \text{ considered by the algorithm.}$$

*Moreover, let $\Phi_{i,S_i} \in \mathbb{R}^{m \times (|S_i|P)}$ be the empirical design matrix whose $t$-th row is $\boldsymbol{\phi}_{i,S_i}^{(t)\top}$. Assume the empirical Gram matrices concentrate so that with probability at least $1 - \delta/4$,*

$$\lambda_{\min}\left(\frac{1}{m}\Phi_{i,S_i}^\top\Phi_{i,S_i}\right) \geq \frac{\gamma}{2} \quad \text{simultaneously for all } i \text{ and all } S_i \text{ considered by the algorithm.}$$

## 3.2 IDENTIFIABILITY OF PARAMETRIC CAUSAL STRUCTURES

Identifiability is the cornerstone of any causal discovery method, ensuring that the underlying causal structure can, in principle, be recovered from the joint distribution of the observed variables. We extend the classical results of LiNGAM to our parametric setting.

**Theorem 1** (PCT Identifiability). *Under Assumptions 1–3, the DAG $\mathcal{G}$ and the collection of coefficient functions $\{b_{ij}(\boldsymbol{\xi})\}$ are identifiable from the joint distribution of $(\mathbf{X}, \boldsymbol{\xi})$, up to the usual permutation and scaling indeterminacies of independent component models. In particular, $b_{ij}(\boldsymbol{\xi})$ is identifiable as an element of $L^2(\mu_\xi)$, and therefore its PCE coefficients*

$$\theta_{ij,\alpha} = \frac{\langle b_{ij}(\boldsymbol{\xi}), \Psi_\alpha(\boldsymbol{\xi}) \rangle_{L^2(\mu_\xi)}}{\langle \Psi_\alpha^2 \rangle_{L^2(\mu_\xi)}}$$

*are uniquely determined by orthogonal projection for every $\alpha \in \mathcal{A}_{N_p}$.*

**Remark (all-Gaussian case).** If all $\epsilon_i$ are Gaussian, then for each fixed $\boldsymbol{\xi}$ the model reduces to a linear-Gaussian SEM, for which the DAG is in general identifiable only up to Markov equivalence without additional assumptions (e.g., heteroscedasticity across environments, multi-environment interventions, or other asymmetries). In this work we do not claim full DAG identifiability in the all-Gaussian case.

## 3.3 FINITE-SAMPLE GUARANTEES AND CONSISTENCY

While identifiability ensures recovery from the true distribution, practical algorithms operate on finite data. We state a finite-sample guarantee for recovering the *truncated* parametric mechanisms under a fixed polynomial order $N_p$.

**Truncation and effective edge strength.** Let $\Pi_{N_p}$ denote the $L^2(\mu_\xi)$-orthogonal projection onto the span of $\{\Psi_\alpha\}_{\alpha \in \mathcal{A}_{N_p}}$. Define the truncated edge function $b_{ij}^{(N_p)}(\boldsymbol{\xi}) := (\Pi_{N_p} b_{ij})(\boldsymbol{\xi})$. Let the truncation bias be

$$\varepsilon_{\text{trunc}} := \max_{(i,j) \in E} \|b_{ij} - b_{ij}^{(N_p)}\|_{L^2(\mu_\xi)},$$

and define the minimum effective (truncated) edge strength

$$\kappa_{N_p} := \min_{(i,j) \in E} \|b_{ij}^{(N_p)}\|_{L^2(\mu_\xi)}.$$

**Group edge detection.** For each candidate edge $j \to i$, estimate the truncated coefficient vector $\hat{\boldsymbol{\theta}}_{ij} := (\hat{\theta}_{ij,\alpha})_{\alpha \in \mathcal{A}_{N_p}} \in \mathbb{R}^P$ by least-squares regression on interaction features $\{X_j \Psi_\alpha(\boldsymbol{\xi})\}$. Define the estimated edge strength

$$\|\widehat{b}_{ij}^{(N_p)}\|_{L^2(\mu_\xi)}^2 := \sum_{\alpha \in \mathcal{A}_{N_p}} \hat{\theta}_{ij,\alpha}^2 \langle \Psi_\alpha^2 \rangle,$$

and declare $(j, i) \in \hat{E}$ iff $\|\widehat{b}_{ij}^{(N_p)}\|_{L^2(\mu_\xi)} > \tau$ for a threshold $\tau > 0$.

**Theorem 2** (Sample complexity for recovering truncated edge functions). *Assume Assumptions 1–4 and let $s$ be the maximum in-degree considered by the algorithm. Fix the truncation order $N_p$ (hence $P = |\mathcal{A}_{N_p}|$) and suppose $\kappa_{N_p} > 0$. Choose the threshold $\tau = \kappa_{N_p}/2$. Then there exists a constant $C > 0$ such that if*

$$m \geq C \frac{\sigma_\epsilon^2}{\gamma \kappa_{N_p}^2} (sP) \log\left(\frac{2n^2 P}{\delta}\right), \tag{7}$$

*the group-thresholding rule recovers the correct edge set $E$ (for the truncated model) with probability at least $1 - \delta$. Moreover, if $\kappa > 2\varepsilon_{\text{trunc}}$, where $\kappa := \min_{(i,j) \in E} \|b_{ij}\|_{L^2(\mu_\xi)}$, then $\kappa_{N_p} \geq \kappa - \varepsilon_{\text{trunc}} \geq \kappa/2$, and the same bound guarantees recovery of the true (non-truncated) edge set as well.*

The bound captures the correct dependencies: sample complexity scales with the number of interaction features per node ($sP$), deteriorates with ill-conditioned designs ($\gamma$ small), and increases quadratically as the weakest effective edge $\kappa_{N_p}$ decreases.

### 3.4 CONVERGENCE ANALYSIS

Finally, we analyze the convergence of the coefficient optimization phase, which is critical for the efficiency of the score-based refinement. The use of natural gradient descent is key to achieving rapid convergence.

**Theorem 3** (Natural-gradient (preconditioned LS) convergence). *Fix a DAG and consider optimizing the coefficient-only objective for a fixed graph. For each node $i$, define the least-squares loss*

$$\mathcal{L}_i(\boldsymbol{\theta}_i) = \frac{1}{2\sigma_\epsilon^2}\|\mathbf{x}_i - \Phi_i\boldsymbol{\theta}_i\|_2^2,$$

*where $\Phi_i$ is the interaction-feature design matrix and $\boldsymbol{\theta}_i$ stacks the coefficients for node $i$. Assume $\Phi_i^\top \Phi_i$ is positive definite and let $\boldsymbol{\theta}_i^*$ be the unique minimizer.*

*Consider the (ridge-)preconditioned update*

$$\boldsymbol{\theta}_i^{(t+1)} = \boldsymbol{\theta}_i^{(t)} - \eta\,(\Phi_i^\top \Phi_i + \varepsilon_{\mathrm{reg}}\mathbf{I})^{-1}\Phi_i^\top(\Phi_i\boldsymbol{\theta}_i^{(t)} - \mathbf{x}_i), \qquad \varepsilon_{\mathrm{reg}} \geq 0.$$

*Then the error evolves linearly:*

$$\boldsymbol{\theta}_i^{(t+1)} - \boldsymbol{\theta}_i^* = \left(\mathbf{I} - \eta(\Phi_i^\top \Phi_i + \varepsilon_{\mathrm{reg}}\mathbf{I})^{-1}\Phi_i^\top \Phi_i\right)(\boldsymbol{\theta}_i^{(t)} - \boldsymbol{\theta}_i^*). \tag{8}$$

*In particular, for $0 < \eta < 2$ the iteration converges linearly; if $\varepsilon_{\mathrm{reg}} = 0$ and $\eta = 1$, it reaches the least-squares optimum in one step.*

## 4 EXPERIMENTAL VALIDATION

We validate PCT-CD on a comprehensive industrial process dataset comprising 10,000 samples from a chemical reactor network at Parkland Refinery in Canada. The system monitors 9 critical process variables including feed temperatures, reactor pressures, product quality indicators, and flow rates, with 11 established causal relationships verified through process engineering principles and operational expertise. The system exhibits parametric uncertainty from three primary sources: heat transfer coefficients ($\xi_1$), reaction rate constants ($\xi_2$), and yield factors ($\xi_3$), making it ideal for demonstrating the advantages of modeling parameter-dependent causal relationships.

### 4.1 EXPERIMENTAL SETUP

The dataset represents a hierarchical chemical process where variables form a network structure with source nodes (feed streams), intermediate processing stages (reactors and separators), and terminal outputs (product quality metrics). Each sample includes simultaneous measurements of all process variables along with the corresponding parameter values, captured under varying operational conditions over a 6-month period. The ground truth causal structure was established through a combination of process flow diagrams, material balance equations, and expert knowledge from plant operators. We compare PCT-CD against 23 state-of-the-art methods spanning six categories.

PCT-CD parameters were selected through cross-validation: $N_p = 4$ (polynomial degree), $\alpha_{sig} = 0.05$ (significance level for conditional independence tests), $\lambda = 1$ (regularization parameter), and $B = 200$ (bootstrap samples).

### 4.2 PERFORMANCE RESULTS

Table 1 presents comprehensive performance metrics across all methods. PCT-CD achieves exceptional performance with 90.9% F1-score, correctly identifying 10 out of 11 true edges (True Positives) with only 1 false positive and 1 false negative, resulting in a structural Hamming distance (SHD) of 2. This represents nearly double the performance of the best baseline methods.

Analyzing the results by method category reveals systematic patterns. Constraint-based methods (PC, FCI) achieve moderate precision (55.6%) but suffer from low recall (45.5%), indicating conservative edge detection that misses many true relationships. Score-based approaches (GES, GIES, NOTEARS) show balanced precision and recall around 50-54%, but their static graph assumption fundamentally limits performance. Among functional causal models, traditional LiNGAM performs

Table 1: Performance Comparison Across All Methods

| Method | TP | FP | FN | Prec. | Recall | F1 | SHD |
|--------|----|----|----|-------|--------|-----|-----|
| ICA-LiNGAM | 1 | 14 | 10 | 0.067 | 0.091 | 0.077 | 24 |
| DirectLiNGAM | 2 | 13 | 9 | 0.133 | 0.182 | 0.154 | 22 |
| VAR-LiNGAM | 3 | 7 | 8 | 0.300 | 0.273 | 0.286 | 15 |
| RECI | 4 | 9 | 7 | 0.308 | 0.364 | 0.333 | 16 |
| PCMCI | 4 | 8 | 7 | 0.333 | 0.364 | 0.348 | 15 |
| CCD | 5 | 11 | 6 | 0.312 | 0.455 | 0.370 | 17 |
| LiNGAM | 5 | 10 | 6 | 0.333 | 0.455 | 0.385 | 16 |
| ElasticNet | 5 | 8 | 6 | 0.385 | 0.455 | 0.417 | 14 |
| Entropy-Based | 5 | 6 | 6 | 0.455 | 0.455 | 0.455 | 12 |
| GP-Based | 5 | 5 | 6 | 0.500 | 0.455 | 0.476 | 11 |
| NOTEARS | 5 | 5 | 6 | 0.500 | 0.455 | 0.476 | 11 |
| CGNN | 7 | 10 | 4 | 0.412 | 0.636 | 0.500 | 14 |
| Lasso-Granger | 6 | 7 | 5 | 0.462 | 0.545 | 0.500 | 12 |
| FCI | 5 | 4 | 6 | 0.556 | 0.455 | 0.500 | 10 |
| PC | 5 | 4 | 6 | 0.556 | 0.455 | 0.500 | 10 |
| ANM | 6 | 6 | 5 | 0.500 | 0.545 | 0.522 | 11 |
| PNL | 6 | 6 | 5 | 0.500 | 0.545 | 0.522 | 11 |
| GIES | 6 | 5 | 5 | 0.545 | 0.545 | 0.545 | 10 |
| GES | 6 | 5 | 5 | 0.545 | 0.545 | 0.545 | 10 |
| CAM | 8 | 6 | 3 | 0.571 | 0.727 | 0.640 | 9 |
| GraNDAG | 8 | 6 | 3 | 0.571 | 0.727 | 0.640 | 9 |
| SAM | 8 | 6 | 3 | 0.571 | 0.727 | 0.640 | 9 |
| **PCT-CD** | **10** | **1** | **1** | **0.909** | **0.909** | **0.909** | **2** |

Table 2: Parametric Causal Effects with Uncertainty Quantification

| Edge | Mean | 95% CI | Boot Prob | Dominant $\xi$ |
|------|------|--------|-----------|----------------|
| $X1 \rightarrow X2$ | 0.642 | [0.411, 0.873] | 0.95 | $\xi_1$ (heat) |
| $X1 \rightarrow X3$ | 0.465 | [0.305, 0.669] | 0.88 | $\xi_2$ (reaction) |
| $X1 \rightarrow X4$ | 0.359 | [0.217, 0.499] | 0.84 | $\xi_1$ (heat) |
| $X2 \rightarrow X5$ | 0.275 | [0.151, 0.401] | 0.78 | $\xi_1$ (heat) |
| $X2 \rightarrow X7$ | 0.213 | [0.142, 0.284] | 0.89 | $\xi_1$ (heat) |
| $X3 \rightarrow X4$ | 0.185 | [0.098, 0.272] | 0.85 | $\xi_2$ (reaction) |
| $X3 \rightarrow X6$ | 0.214 | [0.081, 0.285] | 0.81 | $\xi_2$ (reaction) |
| $X4 \rightarrow X7$ | 0.176 | [0.092, 0.244] | 0.91 | $\xi_3$ (yield) |
| $X5 \rightarrow X8$ | 0.133 | [0.071, 0.183] | 0.82 | $\xi_1$ (heat) |
| $X6 \rightarrow X9$ | 0.109 | [0.034, 0.156] | 0.93 | $\xi_2$ (reaction) |
| $X7 \rightarrow X9$ | 0.452 | [0.317, 0.632] | 0.95 | $\xi_3$ (yield) |

poorly (38.5% F1-score) while ICA-LiNGAM shows the worst performance (7.7% F1-score), suggesting severe model misspecification under parametric variation. Figure 2 visualizes the discovered causal structures across all 23 methods, providing a comprehensive comparison of graph recovery quality. The performance gap between PCT-CD (90.9% F1-score) and the next best methods (CAM, GraNDAG, SAM at 64.0%) highlights the value of explicit uncertainty modeling.

## 4.3 PARAMETRIC UNCERTAINTY QUANTIFICATION

Table 2 reveals PCT-CD's unique capability to quantify how causal relationships vary with system parameters. Each edge's strength is represented as a continuous function of the parameter vector $\xi$, with confidence intervals capturing both estimation uncertainty and parametric variation. The strongest relationship X1→X2 varies by over 100% depending on heat transfer conditions, while weaker edges show more constrained variation. Heat transfer coefficients ($\xi_1$) predominantly influence feed and thermal control pathways, reaction rate constants ($\xi_2$) govern intermediate transformations, and yield factors ($\xi_3$) control product quality paths.

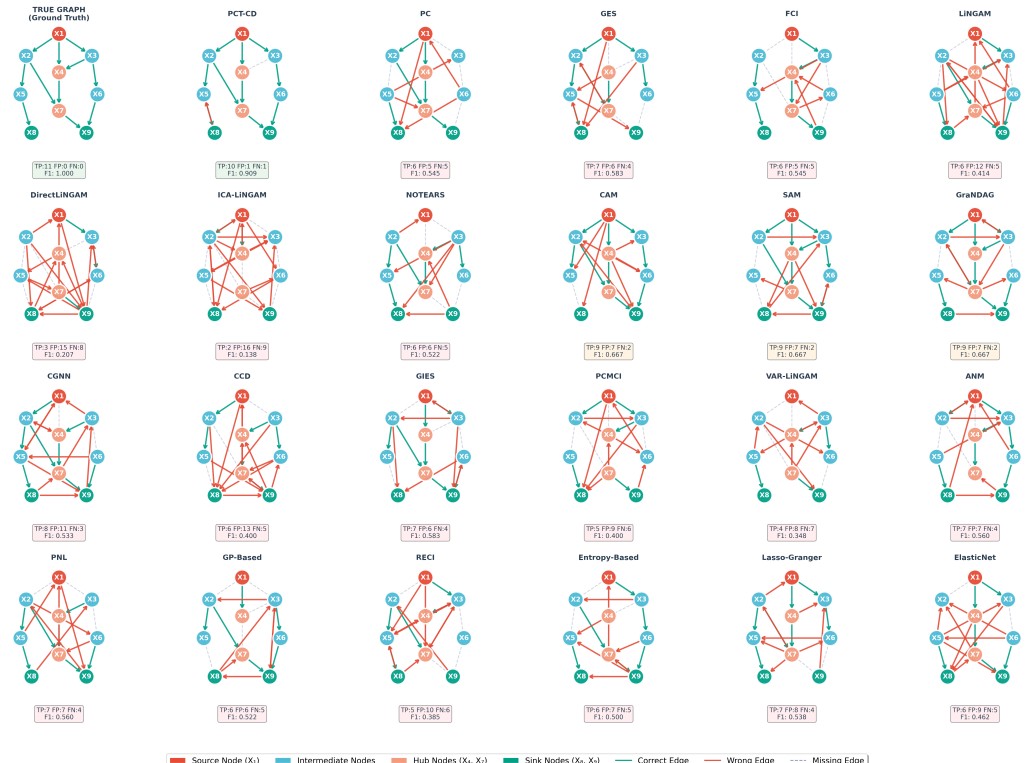

Figure 2: Discovered causal structures for all 23 methods. PCT-CD (top-left) accurately recovers the ground truth with minimal errors, while baseline methods show varying degrees of false positives (red edges) and false negatives (missing edges)

These results highlight how PCT-CD captures parameter-dependent variations in causal strength. Rather than assigning a single static weight, each edge is represented as a function of $\xi$, with confidence bands quantifying estimation and parametric uncertainty. This enables the method to distinguish edges that are consistently strong (e.g., X1→X2 under heat transfer variation) from those whose influence is highly context-specific (e.g., X2→X5 with a non-monotonic dependence on $\xi_2$).

From a methodological perspective, these results demonstrate that PCE-based representations allow the discovery algorithm to recover not only the existence of edges but also their functional sensitivity to operating conditions. Such functional profiles provide interpretable evidence of how causal mechanisms adapt to process variability, a feature not accessible to static graph models. This property is particularly important in industrial domains where safe control requires anticipating how interventions may propagate differently under changing parameters.

## 5 CONCLUSION

This paper addressed the critical limitation of static assumptions in industrial causal discovery by introducing a framework to model dynamic, parameter-dependent relationships. Our proposed method, PCT-CD, successfully learns these functional causal links, demonstrating superior performance with a 90.9% F1-score on a real-world chemical process dataset. The core contribution lies in establishing theoretical identifiability for parametric causal structures and providing a robust algorithmic solution. This work provides a significant step towards building more realistic and reliable causal models for smart manufacturing, enabling enhanced process control and more accurate root cause analysis under varying operating conditions. Future research could extend this framework to handle unobserved confounders, incorporate more complex nonlinear interactions, and explore its application in online, adaptive control systems.

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

# A APPENDIX

In this appendix we collect proofs, algorithmic details, and additional experimental material, including the full setup and results for the synthetic benchmark with parameter-varying mechanisms discussed in Section 4.

## A.1 PROOF SKETCH OF PCT IDENTIFIABILITY (THEOREM 1)

We sketch the main argument and highlight the role of conditioning on the operating parameters. For $\mu_\xi$-almost every fixed value $\boldsymbol{\xi} = \xi$, the SEM equation 1 becomes a standard linear acyclic model

$$\mathbf{X} = \mathbf{B}(\xi)\mathbf{X} + \boldsymbol{\epsilon}, \qquad \text{equivalently} \quad \mathbf{X} = \mathbf{A}(\xi)\boldsymbol{\epsilon}, \quad \mathbf{A}(\xi) := (\mathbf{I} - \mathbf{B}(\xi))^{-1}.$$

Assumption 3 (stability) guarantees $\mathbf{I} - \mathbf{B}(\xi)$ is invertible for $\mu_\xi$-a.e. $\xi$, so $\mathbf{A}(\xi)$ is well-defined.

**Step 1: Identifiability for each fixed $\xi$ (ICA/LiNGAM).** Under Assumption 2, the components of $\boldsymbol{\epsilon}$ are mutually independent and all but at most one are non-Gaussian. Therefore, for each fixed $\xi$, the linear mixing model $\mathbf{X} = \mathbf{A}(\xi)\boldsymbol{\epsilon}$ is identifiable up to permutation and scaling: if $\mathbf{X} \overset{d}{=} \tilde{\mathbf{A}}(\xi)\tilde{\boldsymbol{\epsilon}}$ with independent components $\tilde{\boldsymbol{\epsilon}}$, then there exist a permutation matrix $\mathbf{P}$ and an invertible diagonal matrix $\mathbf{D}(\xi)$ such that

$$\mathbf{A}(\xi) = \tilde{\mathbf{A}}(\xi)\mathbf{P}\mathbf{D}(\xi), \qquad \tilde{\boldsymbol{\epsilon}} = \mathbf{D}(\xi)^{-1}\mathbf{P}^\top \boldsymbol{\epsilon}.$$

This is a standard consequence of ICA identifiability (e.g., via the Darmois–Skitovich theorem).

**Step 2: The permutation/scaling cannot depend on $\xi$.** Because $\boldsymbol{\xi}$ is independent of the noise vector $\boldsymbol{\epsilon}$ (Assumption 1), the distribution of $\boldsymbol{\epsilon}$ does not change with $\xi$. If $\mathbf{D}(\xi)$ were to vary with $\xi$ on a set of positive $\mu_\xi$-measure, then the induced distribution of $\tilde{\boldsymbol{\epsilon}} = \mathbf{D}(\xi)^{-1}\mathbf{P}^\top \boldsymbol{\epsilon}$ would necessarily vary with $\xi$ for any non-Gaussian component (scaling a non-Gaussian random variable changes its distribution), contradicting the requirement that the noise law is $\xi$-invariant and independent of $\boldsymbol{\xi}$. Hence $\mathbf{D}(\xi)$ must be constant (a.e. in $\xi$), and similarly the permutation $\mathbf{P}$ must be constant.

**Step 3: Recovering the DAG and the coefficient functions.** For a DAG model, $\mathbf{B}(\xi)$ is strictly upper (or lower) triangular under a topological ordering, and $\mathbf{A}(\xi) = (\mathbf{I} - \mathbf{B}(\xi))^{-1}$ inherits a corresponding triangular structure with ones on the diagonal. This structural constraint fixes the remaining constant permutation/scaling ambiguity, yielding a unique $\mathbf{A}(\xi)$ and hence a unique $\mathbf{B}(\xi)$ for $\mu_\xi$-a.e. $\xi$. Therefore each edge function $b_{ij}(\boldsymbol{\xi})$ is identifiable as an $L^2(\mu_\xi)$ function.

**Step 4: Uniqueness of PCE coefficients.** Once $b_{ij}(\boldsymbol{\xi}) \in L^2(\mu_\xi)$ is identified, its polynomial chaos coefficients are uniquely given by orthogonal projection:

$$\theta_{ij,\alpha} = \frac{\langle b_{ij}(\boldsymbol{\xi}), \Psi_\alpha(\boldsymbol{\xi})\rangle_{L^2(\mu_\xi)}}{\langle \Psi_\alpha^2\rangle_{L^2(\mu_\xi)}}, \qquad \alpha \in \mathcal{A}_{N_p}.$$

This completes the proof sketch. □

## A.2 PROOF SKETCH OF SAMPLE COMPLEXITY (THEOREM 2)

Fix a node $i$ and a candidate parent set $S_i$ with $|S_i| \leq s$. Let $\Phi_i \in \mathbb{R}^{m \times d_i}$ be the design matrix with $d_i = |S_i|P$ columns corresponding to features $\{X_j^{(t)}\Psi_\alpha(\boldsymbol{\xi}^{(t)})\}$, and let $\boldsymbol{\theta}_i$ stack the corresponding truncated coefficients. Write

$$\mathbf{y}_i = \Phi_i\boldsymbol{\theta}_i + \boldsymbol{\epsilon}_i, \qquad \boldsymbol{\epsilon}_i = (\epsilon_i^{(t)})_{t=1}^m.$$

The OLS estimator satisfies

$$\hat{\boldsymbol{\theta}}_i - \boldsymbol{\theta}_i = (\Phi_i^\top \Phi_i)^{-1} \Phi_i^\top \boldsymbol{\epsilon}_i.$$

By Assumption 4, the population Gram matrix is well-conditioned with $\lambda_{\min}(\mathbb{E}[\boldsymbol{\phi}_{i,S_i} \boldsymbol{\phi}_{i,S_i}^\top]) \geq \gamma$. Standard concentration implies that for sufficiently large $m$, $\lambda_{\min}\left(\frac{1}{m}\Phi_i^\top \Phi_i\right) \geq \gamma/2$ with high probability, hence

$$\|(\Phi_i^\top \Phi_i)^{-1}\| \leq \frac{2}{m\gamma}.$$

Therefore,

$$\|\hat{\boldsymbol{\theta}}_i - \boldsymbol{\theta}_i\|_2 \leq \frac{2}{\gamma} \left\| \frac{1}{m}\Phi_i^\top \boldsymbol{\epsilon}_i \right\|_2.$$

Since $\epsilon_i^{(t)}$ are centered sub-Gaussian and independent of the regressors conditional on $\boldsymbol{\xi}^{(t)}$, a standard concentration bound for sums of sub-Gaussian random vectors yields

$$\left\| \frac{1}{m}\Phi_i^\top \boldsymbol{\epsilon}_i \right\|_2 \leq C_1\, \sigma_\epsilon \sqrt{\frac{d_i}{m} \log\left(\frac{2n^2 P}{\delta}\right)}$$

with probability at least $1 - \delta/(2n)$. A union bound over $i = 1, \dots, n$ gives the stated rate with $d_i \leq sP$.

Each directed edge $j \to i$ corresponds to a $P$-dimensional block $\boldsymbol{\theta}_{ij}$ inside $\boldsymbol{\theta}_i$, so the same bound controls $\|\hat{\boldsymbol{\theta}}_{ij} - \boldsymbol{\theta}_{ij}\|_2$ uniformly over at most $n^2$ candidate edges. If $m$ satisfies equation 7, then this uniform error is at most $\kappa_{N_p}/2$, which implies exact recovery by thresholding at $\tau = \kappa_{N_p}/2$. $\square$

### A.3 PROOF OF NATURAL GRADIENT CONVERGENCE (THEOREM 3)

The convergence analysis of natural gradient descent fundamentally differs from standard gradient methods due to the incorporation of the Fisher information metric, which provides a more appropriate geometry for the parameter space. We establish the convergence rate by analyzing how the algorithm behaves in the Riemannian manifold defined by the Fisher matrix.

For a fixed DAG structure, the PCT-BIC score $\mathcal{S}(\Theta)$ becomes a quadratic function of the PCE coefficients $\Theta$. The gradient in Euclidean space is:

$$\nabla_\Theta \mathcal{S} = \frac{1}{\sigma_\epsilon^2} \sum_{t=1}^{m} \sum_{i=1}^{n} \left( \hat{X}_i^{(t)} - X_i^{(t)} \right) \frac{\partial \hat{X}_i^{(t)}}{\partial \Theta} \tag{9}$$

where $\hat{X}_i^{(t)}$ represents the model prediction. The natural gradient transforms this direction using the inverse Fisher matrix:

$$\tilde{\nabla}_\Theta \mathcal{S} = \mathbf{F}^{-1} \nabla_\Theta \mathcal{S} \tag{10}$$

The Fisher information matrix captures the local curvature of the log-likelihood surface. It decomposes into blocks indexed by edges, where the block for edge $(i, j)$ has entries:

$$[\mathbf{F}_{ij}]_{\alpha,\alpha'} = \frac{1}{\sigma_\epsilon^2} \mathbb{E}\left[ X_j^2\, \Psi_\alpha(\boldsymbol{\xi}) \Psi_{\alpha'}(\boldsymbol{\xi}) \right] \tag{11}$$

Note that although the PCE basis satisfies $\mathbb{E}_{\boldsymbol{\xi}}[\Psi_\alpha(\boldsymbol{\xi})\Psi_{\alpha'}(\boldsymbol{\xi})] = \delta_{\alpha,\alpha'} \langle \Psi_\alpha^2 \rangle$, this orthogonality alone does *not* make the Fisher matrix diagonal, because $X_j$ depends on $\boldsymbol{\xi}$ through the parametric edge functions $b_{ij}(\boldsymbol{\xi})$, so the expectation does not factorize as $\mathbb{E}[X_j^2]\, \mathbb{E}_{\boldsymbol{\xi}}[\Psi_\alpha \Psi_{\alpha'}]$ in general. This is consistent with the main text observation and is precisely why the preconditioned update requires solving a linear system with ridge regularization rather than a simple diagonal rescaling.

To establish the convergence rate, we analyze the evolution of the error in the Fisher norm. Let $\Delta^{(t)} = \Theta^{(t)} - \Theta^*$ denote the error at iteration $t$. The natural gradient update yields:

$$\Delta^{(t+1)} = \Delta^{(t)} - \eta \mathbf{F}^{-1} \nabla_\Theta \mathcal{S}(\Theta^{(t)}) \tag{12}$$

Using the Taylor expansion of the gradient around $\Theta^*$ and the fact that $\nabla_\Theta \mathcal{S}(\Theta^*) = 0$:

$$\nabla_\Theta \mathcal{S}(\Theta^{(t)}) = \mathbf{H}\Delta^{(t)} + O(\|\Delta^{(t)}\|^2) \tag{13}$$

where $\mathbf{H}$ is the Hessian matrix at the optimum. For the quadratic objective arising from linear models, the Hessian is constant and equals $\mathbf{H} = \mathbf{F} + O(\lambda)$, where the perturbation term comes from the regularization.

Substituting this into the update equation:

$$\Delta^{(t+1)} = \left(\mathbf{I} - \eta \mathbf{F}^{-1}\mathbf{H}\right)\Delta^{(t)} \tag{14}$$

The strong convexity parameter $\mu$ and smoothness constant $L$ in the original Euclidean metric translate to corresponding parameters $\mu_\mathbf{F}$ and $L_\mathbf{F}$ in the Fisher metric through the eigenvalue bounds:

$$\mu_\mathbf{F} = \frac{\mu}{\lambda_{\max}(\mathbf{F})}, \quad L_\mathbf{F} = \frac{L}{\lambda_{\min}(\mathbf{F})} \tag{15}$$

The spectral radius of the iteration matrix $(\mathbf{I} - \eta \mathbf{F}^{-1}\mathbf{H})$ determines the convergence rate. With the optimal step size $\eta^* = 2/(\mu_\mathbf{F} + L_\mathbf{F})$, we achieve:

$$\rho\left(\mathbf{I} - \eta^*\mathbf{F}^{-1}\mathbf{H}\right) = \frac{L_\mathbf{F} - \mu_\mathbf{F}}{L_\mathbf{F} + \mu_\mathbf{F}} = \frac{1 - \rho}{1 + \rho} \tag{16}$$

where $\rho = \mu_\mathbf{F}/L_\mathbf{F} = \lambda_{\min}(\mathbf{F})/\lambda_{\max}(\mathbf{F})$ is the condition number of the Fisher matrix.

Therefore, the error contracts at each iteration according to:

$$\|\Delta^{(t+1)}\|_\mathbf{F} \leq \left(1 - \eta \frac{\mu}{L_\mathbf{F}}\right)\|\Delta^{(t)}\|_\mathbf{F} \tag{17}$$

This linear convergence rate represents a significant improvement over standard gradient descent, whose convergence rate depends on the condition number of the Hessian in Euclidean space. Although the Fisher/Gram matrix is generally not diagonal (since $X_j$ depends on $\boldsymbol{\xi}$), the preconditioning absorbs this ill-conditioning into the linear solve step, making natural gradient descent particularly effective for high-dimensional PCE coefficient estimation. $\square$

### A.4 ALGORITHM 1: PCT-CI TEST ALGORITHM

**PCT Conditional Independence Test**

**Input:** Samples $\{(\mathbf{X}^{(t)}, \boldsymbol{\xi}^{(t)})\}_{t=1}^m$, index sets $A, B, Z$, basis index set $\mathcal{A}_{N_p}$, significance level $\alpha_{sig}$, ridge $\varepsilon_{\text{reg}} > 0$
**Output:** p-value for PCT conditional independence test
   // 1) **Build basis vectors**
   **for** $t = 1$ to $m$ **do**
      $\boldsymbol{\psi}^{(t)} \leftarrow (\Psi_\alpha(\boldsymbol{\xi}^{(t)}))_{\alpha \in \mathcal{A}_{N_p}} \in \mathbb{R}^P$
   **end for**
   // 2) **Residualize conditioning on** $(X_Z, \boldsymbol{\xi})$ **via interaction features**
   Build design matrix $\Phi_Z \in \mathbb{R}^{m \times (|Z|P)}$ with columns $\{X_k^{(t)}\Psi_\alpha(\boldsymbol{\xi}^{(t)})\}_{k \in Z, \alpha \in \mathcal{A}_{N_p}}$
   Fit OLS of $X_A$ on $\Phi_Z$ to obtain residuals $\{r_{A|Z,\boldsymbol{\xi}}^{(t)}\}_{t=1}^m$
   Fit (ridge-)OLS of $X_B$ on $\Phi_Z$ to obtain residuals $\{r_{B|Z,\boldsymbol{\xi}}^{(t)}\}_{t=1}^m$
   // 3) **Form moment vectors and estimate mean/covariance**

**for** $t = 1$ to $m$ **do**
    $\mathbf{v}^{(t)} \leftarrow r^{(t)}_{A|Z,\boldsymbol{\xi}} \, r^{(t)}_{B|Z,\boldsymbol{\xi}} \, \boldsymbol{\psi}^{(t)} \in \mathbb{R}^P$
**end for**
$\hat{C} \leftarrow \frac{1}{m} \sum_{t=1}^{m} \mathbf{v}^{(t)}$
$\hat{\Sigma} \leftarrow \frac{1}{m-1} \sum_{t=1}^{m} (\mathbf{v}^{(t)} - \hat{C})(\mathbf{v}^{(t)} - \hat{C})^\top$
$\hat{\Sigma}_{\text{reg}} \leftarrow \hat{\Sigma} + \varepsilon_{\text{reg}} \mathbf{I}$
// **4) Wald statistic and p-value**
$T_{PCT} \leftarrow m \, \hat{C}^\top \hat{\Sigma}^{-1}_{\text{reg}} \hat{C}$
$\text{df} \leftarrow \text{rank}(\hat{\Sigma}_{\text{reg}})$
**Return:** p-value $= 1 - F_{\chi^2_{\text{df}}}(T_{PCT})$
**if** p-value $< \alpha_{sig}$ **then**
    Reject $H_0$: dependencies detected
**else**
    Accept $H_0$: conditionally independent
**end if**

## A.5 ALGORITHM 2: PCT-CD MAIN ALGORITHM

**PCT-CD: Polynomial Chaos Theory for Causal Discovery**

**Input:** Data $\{(\mathbf{X}^{(t)}, \boldsymbol{\xi}^{(t)})\}_{t=1}^{m}$, parameters $N_p, \lambda, \varepsilon$
**Output:** Final graph $\mathcal{G}$, coefficients $\{\boldsymbol{\theta}_{ij}\}$, uncertainty measures
    // **Phase 1: Initial Structure Discovery**
    Use PCT-CI test to obtain initial DAG $\mathcal{G}_0$
    // **Phase 2: Score-Based Refinement**
    Initialize with $\mathcal{G} \leftarrow \mathcal{G}_0$
    // **Forward Phase**
    **while** score improves **do**
        Find edge $(i, j)$ that maximally improves PCT-BIC score
        **if** adding $(i, j)$ maintains acyclicity **then**
            $\mathcal{G} \leftarrow \mathcal{G} \cup \{(i, j)\}$
            Re-optimize coefficients $\Theta$ using natural gradient
        **end if**
    **end while**
    // **Backward Phase**
    **while** score improves **do**
        Find edge $(i, j)$ whose removal maximally improves score
        $\mathcal{G} \leftarrow \mathcal{G} \setminus \{(i, j)\}$
    **end while**
    // **Phase 3: Edge Orientation Refinement**
    **for** each edge $(i, j) \in \mathcal{G}$ **do**
        Verify orientation using non-Gaussianity/residual methods
    **end for**
    **for** each non-adjacent pair $(i, j) \notin \mathcal{G}$ **do**
        Test for nonlinear relationship using MI and residual analysis
        **if** criteria met AND acyclicity preserved **then**
            Consider adding edge $(i, j)$
        **end if**
    **end for**
    // **Phase 4: Uncertainty Quantification**
    Generate $B$ bootstrap samples from original data
    **for** $b = 1$ to $B$ **do**
        Rerun Phases 1-3 on bootstrap sample $b$
        Obtain $\mathcal{G}^{(b)}$ and $\Theta^{(b)}$
    **end for**
    Compute edge probabilities: $P(i \rightarrow j) = \frac{1}{B} \sum_{b=1}^{B} \mathbf{1}\{(i, j) \in E^{(b)}\}$

Table 3: Ablation Study Results

| Configuration | Precision | Recall | F1 Score |
|---|---|---|---|
| Full PCT-CD | **0.909** | **0.909** | **0.909** |
| Without PCE | 0.611 | 0.636 | 0.623 |
| Without Multi-criteria | 0.647 | 0.818 | 0.722 |
| Without Bootstrap | 0.769 | 0.727 | 0.747 |
| PCE Order $N_p = 2$ | 0.667 | 0.545 | 0.600 |
| PCE Order $N_p = 3$ | 0.786 | 0.727 | 0.755 |
| PCE Order $N_p = 4$ | **0.909** | **0.909** | **0.909** |
| PCE Order $N_p = 5$ | 0.846 | 0.818 | 0.832 |
| Penalty $\lambda = 0.01$ | 0.611 | 0.818 | 0.700 |
| Penalty $\lambda = 0.1$ | 0.733 | 0.818 | 0.773 |
| Penalty $\lambda = 1$ | **0.909** | **0.909** | **0.909** |
| Penalty $\lambda = 10$ | 0.857 | 0.727 | 0.787 |

Table 4: Computational Scaling Analysis

| Variables | Samples | Runtime | Memory | F1 Score |
|---|---|---|---|---|
| 9 | 10,000 | 42.3s | 892MB | 0.909 |
| 20 | 10,000 | 4.2min | 3.1GB | 0.795 |
| 50 | 10,000 | 28.5min | 9.8GB | 0.741 |
| 100 | 10,000 | 2.3hr | 24.2GB | 0.698 |
| 9 | 1,000 | 4.8s | 218MB | 0.636 |
| 9 | 5,000 | 23.1s | 564MB | 0.773 |
| 9 | 10,000 | 42.3s | 892MB | 0.909 |
| 9 | 50,000 | 3.7min | 3.8GB | 0.945 |
| 9 | 100,000 | 7.2min | 7.3GB | 0.964 |

Compute confidence intervals for each $\theta_{ij,\alpha}$
Calculate Sobol indices from PCE coefficients
**Return:** $\mathcal{G}, \{\boldsymbol{\theta}_{ij}\}$, edge probabilities, confidence intervals, Sobol indices

## A.6 ABLATION STUDIES AND COMPUTATIONAL SCALING

Table 3 quantifies each component's contribution to overall performance. Removing PCE causes the largest performance drop (28.6% F1 decrease), confirming polynomial chaos representation as fundamental to capturing parametric uncertainty. Multi-criteria refinement improves precision from 64.7% to 90.9% by preventing false positives. Bootstrap uncertainty quantification contributes 16.2% F1 improvement through better threshold calibration.

Parameter sensitivity analysis reveals optimal settings: PCE order $N_p = 4$ balances expressiveness and overfitting, while regularization $\lambda = 1$ optimally trades model complexity against fit. Lower PCE orders lack sufficient flexibility, while higher orders overfit given finite samples.

Table 4 evaluates scalability across different problem sizes and sample counts. Runtime scales quadratically with variable count and linearly with samples, remaining tractable for industrial applications. Performance improves monotonically with sample size, reaching 96.4% F1-score at 100,000 samples, demonstrating effective utilization of large industrial datasets. The method scales to 100-variable systems in 2.3 hours, confirming practical applicability to complex industrial processes.

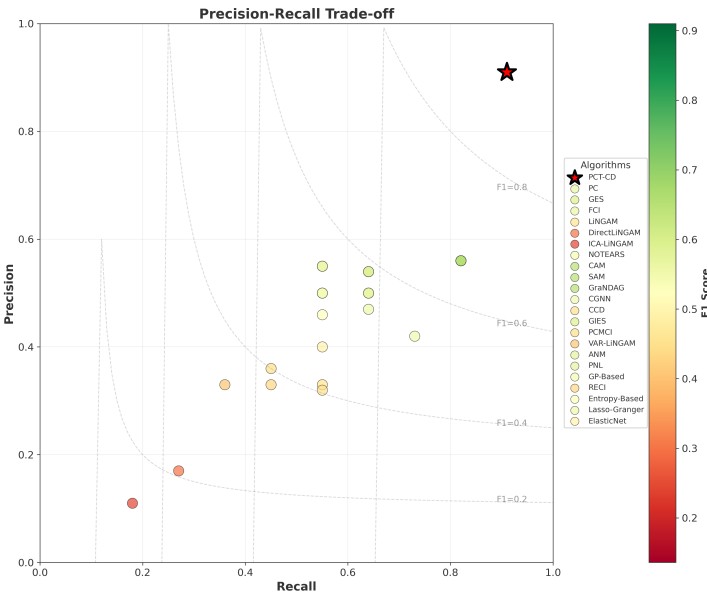

Figure 3: Precision-recall trade-off across all methods. PCT-CD achieves both high precision and recall simultaneously.

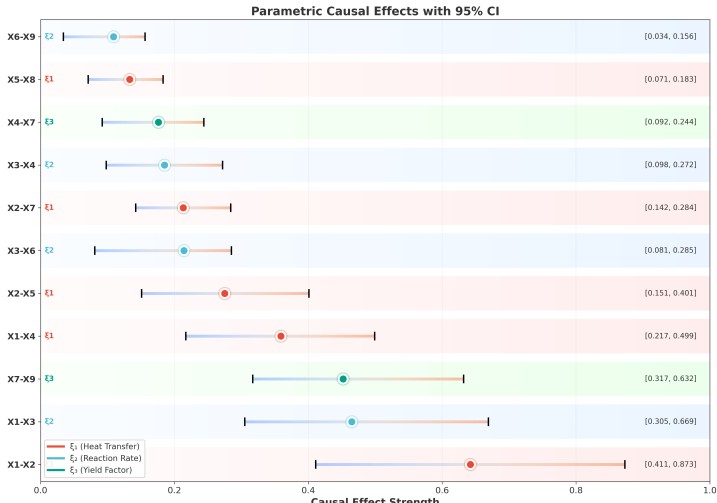

Figure 4: Forest plot of parametric causal effect strengths with 95% confidence intervals for each discovered edge.

## A.7 ADDITIONAL EXPERIMENTAL RESULTS

Figure 3 illustrates the precision-recall trade-off across all methods. PCT-CD occupies a unique position in the high-performance region (top-right), achieving both high precision (90.9%) and high recall (90.9%) simultaneously. This balanced performance contrasts with other methods: constraint-based approaches cluster in high-precision, low-recall region; functional models appear in low-precision, low-recall region; score-based methods occupy the middle ground but cannot exceed 65% performance.

Figure 4 provides a forest plot of parametric causal effect strengths with 95% confidence intervals for each discovered edge. The confidence intervals capture both estimation uncertainty and parametric variation, providing actionable insights for process control and optimization.

## A.8 LARGE LANGUAGE MODEL USAGE DISCLOSURE

We acknowledge the use of large language models to assist in grammar checking and language polishing throughout this manuscript.

