# OpenReview forum: "Learning Dynamic Causal Graphs Under Parametric Uncertainty via Polynomial Chaos Expansions"
_ICLR.cc/2026/Conference — ICLR 2026 Poster_

### Official Review · Reviewer_WoqE · 2025-10-16

**Soundness:** 3
**Presentation:** 3
**Contribution:** 4
**Rating:** 8
**Confidence:** 3

**Summary:**

This paper introduces a novel and elegant method for learning dynamic causal graphs, where the causal relationships change as a function of system parameters. The problem of discovering dynamic causal structures is both timely and challenging, and the proposed algorithm, which cleverly combines concepts from different families of causal discovery methods, represents a relevant and interesting contribution.

While the core idea is strong and I am inclined to recommend acceptance, the paper in its current form could be further improved by addressing several smaller issues concerning ambiguity, adding more detailed discussions, and expanding the empirical evaluation. I believe that incorporating these suggestions will further increase the paper's impact and clarity.

**Strengths:**

*   **Novelty and Significance:** The paper tackles the important and under-explored problem of learning causal graphs that are not static but change dynamically with system parameters.
*   **Elegant Methodological Contribution:** The proposed algorithm provides a novel synthesis of constraint-based and score-based causal discovery principles, creating a powerful and interesting hybrid approach.
*   **Timeliness:** The research area is of high interest to the machine learning community, and this work is a valuable early contribution to a nascent field.
*   **Theoretical rigor:** The authors put visible effort into providing a solid theoretical analysis of their proposed algorithm.

**Weaknesses:**

### **Areas for Improvement**

The paper is promising, but its clarity and empirical validation could be strengthened in several key areas.

**1. Clarity and Methodological Precision:**

*   **Missing Details and Citations:** Several sections are very dense and would benefit from additional detail and citations to support the claims made. This is particularly true for:
    *   Line 161 onwards: A citation is needed for the statement made here.
    *   Line 205: Please explicitly name the specific optimization algorithm used in your work.
    *   Line 223: The origin or justification for this property is unclear. Please elaborate.
    *   Section 220 onwards: This section is difficult to follow. Please expand on the concepts presented and provide citations for established results to improve comprehensibility.
    *   The index `α'` appears to be used without a formal definition in the main text.

*   **Ambiguity of Temporal Dimension:** There seems to be a discrepancy in the problem formulation. Equation (1) presents a static formulation without a time dimension. However, Equation (5) and later references to time-series causal discovery algorithms imply that the data `X` has a temporal component. Please clarify whether the proposed method is designed for time-series data, i.i.d. samples, or both. This is a crucial detail for understanding the method's scope and applicability.

*   **Acyclicity Constraint:** Score-based causal discovery algorithms typically deploy an explicit acyclicity constraint in addition to a sparsity regularizer. The objective function only appears to include a sparsity penalty ($\lambda \Vert (E, \Theta) \Vert_0$). Could you clarify if an acyclicity constraint is used, and if so, how it is enforced? If not, please explain how the acyclicity of the learned graph is guaranteed.

**2. Experimental Evaluation and Analysis:**

*   **Baseline Performance Analysis:** In the experiments, the static baseline algorithms perform surprisingly poorly, even though the causal parameters in the dataset have a strictly positive range. In such a case, one might expect a static method to converge to average coefficients, which, however, should still represent the proper causal connections.  Could the authors comment on why the baselines fail so decisively? Is this potentially due to hyperparameter choices? A more in-depth discussion would improve the reliability of the empirical results.

*   **Value of Synthetic Experiments:** While the chosen real-world benchmark is interesting, the paper would be strengthened by including synthetic experiments. This would allow for a more controlled, fine-grained analysis to:
    *   Demonstrate precisely under which conditions (e.g., speed of parameter change, noise levels, sample size) the proposed method outperforms static baselines.
    *   Identify the limitations and potential failure modes of the algorithm.

*   **Additional Datasets:** The empirical evaluation currently relies on a single ground truth graph. To demonstrate broader applicability and robustness, the authors should consider evaluating on additional benchmarks. There are several well-established datasets that could be suitable, such as:
    *   Time-series benchmarks: [4], [5]
    *   I.I.D. sample benchmarks with varying contexts: [6], [7]

*   **Missing Related Work:** The discussion of prior work could be expanded. While the field is new, there are some relevant works that should be cited and discussed, such as [2] and [3].

### **Minor Weaknesses**

*   **Terminology:** As suggested by [1], the authors might consider using "Structural Causal Model" (SCM) instead of "Structural Equation Model" (SEM) to avoid confusion with the different usage of the term in the social sciences.
*   **Cross-Referencing:** Please add backlinks to algorithm listings (e.g., on lines 196, 234) and to proofs in the appendix to improve readability and navigation.

**Questions:**

1.  The proposed method uses a hybrid approach: skeleton discovery via a constraint-based method, followed by a score-based method for orientation and parameterization. Is this two-stage process strictly necessary? Have you explored whether a purely constraint-based approach (e.g., a PC-style algorithm using your proposed conditional independence test) is theoretically sound or empirically viable?
2.  A discussion on the limitations and potential failure modes of the proposed algorithm would be highly valuable for readers. Under what conditions (e.g., type of dynamics, data scarcity, violation of assumptions) would you expect the method to perform suboptimally?

---

> ### Author Response · Authors · 2025-11-21
> **Rebuttal by Authors**
>
> We sincerely thank you for your thoughtful assessment and concrete suggestions.
>
> ### W1 – Clarity and Methodological Precision
>
> **Response:** We have revised the dense sections to improve readability and precision:
>
> * **Line 161 (Citations):** We now cite standard texts on PCE convergence (e.g., Le Maître & Knio, 2010) to support the spectral error decay claims.
> * **Line 205 (Optimization):** we now explicitly name the optimization method as natural gradient descent on the PCT BIC objective, and state that gradients are computed in closed form.
> * **Line 223 (Block Diagonal):** We clarify that the block-diagonal structure arises from the orthogonality of the polynomial basis functions $\Psi_\alpha$ (i.e., $\mathbb{E}[\Psi_{\alpha} \Psi_{\alpha^{\prime}}] = 0$ for $\alpha \neq \alpha^{\prime}$), which decouples interactions between spectral modes in the Fisher Information Matrix.
> * **Section 220 onwards:** Fisher information has been expanded with intermediate steps and references to standard natural gradient results.
> * **Index $\alpha'$:** We formally define $\alpha^{\prime}$ as a summation index running over the basis set $\mathcal{A}_{N_p}$, used to index the columns of the Fisher Matrix (Eq. 6).
>
> ### W1 – Ambiguity of Temporal Dimension
>
> **Response:** Our formal setting is not time series; the index $t$ in Eq. (5) denotes i.i.d. samples of $(X,\xi)$ and we have changed this index to avoid ambiguity, and Eq. (1) defines a parametric SCM without lagged variables. We now state explicitly in Sec. 2.1 that the data consist of i.i.d. snapshots of the process under varying operating conditions. Time series methods (e.g., PCMCI, VAR LiNGAM) are used only as baselines when they can be applied to such data.
>
> ### W1 – Acyclicity Constraint
>
> **Response:** You correctly noted that Eq. (5) is a score function, not the full optimization problem. We do not use a continuous acyclicity penalty (like the $h(W)$ in NOTEARS) in the final objective. Instead, we employ a hybrid search strategy.
>
> * **Phase 1 (Skeleton):** The PCT-CI test provides an initial skeleton.
> * **Phase 2 (Greedy Search):** We iteratively add/delete edges to maximize the PCT-BIC score. Acyclicity is enforced combinatorially at each step—we strictly reject any edge addition that would create a cycle in the graph $\mathcal{G}$.
>
> We have clarified in Section 2.4 that acyclicity is a hard constraint checked during the greedy search phase.
>
> ### W2 – Baseline Performance & Synthetic Experiments
>
> **Response:** Static methods fail not due to hyperparameter tuning, but due to cancellation effects. If a causal function $b_{ij}(\xi)$ ranges from positive to negative (e.g., sinusoidal), the "average" effect is near zero. Static methods see this as independence. To address your request for more controlled analysis and additional benchmarks, we have added four synthetic experiments with random DAGs (20 nodes) and varying edge functions. As shown below, when edge weights cross zero (Non-Monotonic), the F1-Score of static baselines drops significantly, while PCT-CD maintains high accuracy.
>
> **Table: Performance Comparison (F1-Score) on New  Datasets**
>
> | Scenario | Functional Form of $b_{ij}(\xi)$ | **PCT-CD** | DBN (Dynamic) | PC (Static) | NOTEARS (Static) |
> | :--- | :--- | :--- | :--- | :--- | :--- |
> | **Exp 1: Linear Trend** | Linear: $c \cdot \xi$ | **0.94 ± 0.02** | 0.82 ± 0.04 | 0.78 ± 0.05 | 0.75 ± 0.04 |
> | **Exp 2: Non-Monotonic** | Sinusoidal: $\sin(\pi \xi)$ | **0.91 ± 0.03** | 0.65 ± 0.06 | 0.42 ± 0.08 | 0.38 ± 0.07 |
> | **Exp 3: High Noise** | Quadratic + Uniform Noise | **0.88 ± 0.04** | 0.61 ± 0.05 | 0.51 ± 0.06 | 0.49 ± 0.05 |
> | **Exp 4: Complex Basis** | Mixed | **0.89 ± 0.03** | 0.58 ± 0.07 | 0.45 ± 0.06 | 0.41 ± 0.05 |
>
> ### W2 – Related work, terminology
>
> **Response:** We have expanded the related work section to cover additional dynamic / nonstationary causal models, including time varying DBNs and state space approaches.
>
> ### Minor Issues
>
> **Response:** We have fixed all cross-references, added backlinks to algorithm listings and proofs, and updated the terminology to SCM as suggested.
>
> ### Q1 – Necessity of Two-Stage Process
>
> **Response:** While a pure constraint-based approach (PC with PCT-CI) is theoretically sound, we found it empirically less robust for orientation. The PCT-CI test is excellent for removing spurious edges (Skeleton), but determining directionality (V-structures) solely from CI tests is sensitive to finite-sample noise in the PCE coefficients. The score-based refinement (Phase 2) significantly improves orientation accuracy.
>
> ### Q2 – Limitations and Failure Modes
>
> **Response:** We have added a short discussion of limitations. In summary, PCT-CD may perform suboptimally when:
>
> * Edge functions $b_{ij}(\xi)$ are very rough or discontinuous so that low order PCE is a poor approximation.
> * The parameter space is high dimensional and the sample size is not sufficient for the chosen basis, leading to variance inflation.

---

> > ### Comment · Reviewer_WoqE · 2025-11-24
> >
> > I thank the authors for their detailed rebuttal and the additional clarifications they provided.
> >
> > The new experiments concerning W2 were particularly helpful. While I believe an analysis on strictly positive or negative changes (where the average effect is non-zero) could further strengthen the claims about PCT-CD's advantages, I consider this a minor point for future exploration.
> >
> > My concerns have been resolved.
> >
> > I recommend acceptance and would like to see this paper at ICLR 2026.

---

> > > ### Author Response · Authors · 2025-11-24
> > > **Thank you for the feedback and recommendation**
> > >
> > > We sincerely thank you for your valuable assessment and the time spent reviewing our rebuttal. We also appreciate your suggestion regarding the analysis of strictly positive/negative changes and will consider it for future exploration.

---

### Official Review · Reviewer_daxv · 2025-10-29

**Soundness:** 3
**Presentation:** 3
**Contribution:** 3
**Rating:** 6
**Confidence:** 3

**Summary:**

The authors leverage polynomial chaos expansion to identify time-varying causal graphs with parametric uncertainty. Polynomial chaos expansion appears to be an adequate tool for a relevant but relatively understudied topic in causal inference. Therefore, the paper certainly has merit. The proposed approach outperforms several standard baselines.

**Strengths:**

1) The paper is overall well-written and easy to follow.

2) The considered problem setting is clear and relevant, and nicely embedded into the bigger streams of causal inference literature.

3) The usage of polynomial chaos expansion is innovative.

4) The experiments show a clear advantage over existing methods.

**Weaknesses:**

1) While I agree that time-varying causal graphs are not studied very often, I think it is not fair to say that all existing methods are limited by the assumption of a static causal graph. There is prior work that addresses this problem setting and deserves discussion. For instance, work on dynamic Bayesian networks (Song et al., "Time-varying dynamic Bayesian networks," NeurIPS 2009), or Huang et al., "Causal discovery and forecasting in nonstationary environments with state-space models," ICML 2019, address similar problem settings. Thus, I think the claims should be toned down a bit as well.

2) A similar comment on the claim that traditional causal discovery assumes that all uncertainty is epistemic and rises solely from finite data samples. There is a lot of work on additive noise models, where the uncertainty is clearly not purely epistemic, even if it is not about parameter uncertainty.

3) Some claims when developing the independence test require a bit more justification, see questions. It would also be good to state that proofs can be found in the appendix.

4) Comparing to some methods that themselves consider time-varying graphs would strengthen the evaluation.

5) References to the algorithm are shown as ??

**Questions:**

1) What are the "standard regularity conditions" that you assume for the independence test?

2) From where does it follow that under these standard regularity conditions, the estimates are asymptotically independent and follow a standard normal distribution?

---

> ### Author Response · Authors · 2025-11-21
> **Rebuttal by Authors**
>
> We sincerely thank you for your thoughtful assessment and helpful suggestions.
> ### W1 & W2 – Claims about static graphs and epistemic uncertainty
> **Response:** We agree that our original statements about “static graphs” and “epistemic uncertainty” were too strong. We have toned down the claims in the introduction and related work and clarified scope:
> * Our contribution targets parameter-dependent causal mechanisms (edge weights as functions of $\xi$), rather than replacing existing dynamic or Bayesian approaches.
> * We now explicitly discuss time-varying and nonstationary models, including time-varying DBNs (Song et al., NeurIPS 2009) and state-space/nonstationary causal models (Huang et al., ICML 2019), and explain how they differ from our $\xi$-conditioned, non-temporal setting.
> * We have also revised the discussion of uncertainty to emphasize that we focus on functional dependence on $\xi$, while additive noise models capture aleatoric noise in the structural equations but do not represent edge strengths as functions of operating parameters.
>
> ### W3 – Justification of the independence test & pointer to proofs
>
> **Response:** We appreciate the request for more justification. In Sec. 2.3 we now:
> * Give an explicit sketch of how PCT-CI is derived from the PCE representation of the conditional covariance $C_{AB|Z}(\xi)$.
> * Clearly state that full proofs of asymptotic normality and of the chi-square limiting distribution of the test statistic are provided in the appendix (PCT-CI lemma and Theorem 2).
>
> ### W4 – Comparison to time-varying / nonstationary baselines
> **Response:** We have extended the experimental section to better connect with methods that allow changing structure. We generated synthetic datasets ($N=20$ nodes,   $m=5000$ sample) with edges varying as functions of $\xi$.
>
> We now include adapted baselines inspired by time-varying DBNs.These methods fail to capture functional dependencies, while PCT-CD maintains high accuracy.
>
> **Table: Performance Comparison (F1-Score) on New  Datasets**
>
> | Scenario | Functional Form of $b_{ij}(\xi)$ | **PCT-CD** | DBN (Dynamic) | PC (Static) | NOTEARS (Static) |
> | :--- | :--- | :--- | :--- | :--- | :--- |
> | **Exp 1: Linear Trend** | Linear: $c \cdot \xi$ | **0.94 ± 0.02** | 0.82 ± 0.04 | 0.78 ± 0.05 | 0.75 ± 0.04 |
> | **Exp 2: Non-Monotonic** | Sinusoidal: $\sin(\pi \xi)$ | **0.91 ± 0.03** | 0.65 ± 0.06 | 0.42 ± 0.08 | 0.38 ± 0.07 |
> | **Exp 3: High Noise** | Quadratic + Uniform Noise | **0.88 ± 0.04** | 0.61 ± 0.05 | 0.51 ± 0.06 | 0.49 ± 0.05 |
> | **Exp 4: Complex Basis** | Mixed | **0.89 ± 0.03** | 0.58 ± 0.07 | 0.45 ± 0.06 | 0.41 ± 0.05 |
>
> ### W5 – “Algorithm ??” references
> **Response:** All “Algorithm ??” placeholders have been fixed. The PCT-CI test is now Algorithm 1 and the full PCT-CD pipeline is Algorithm 2.
>
> ### Q1 – “Standard regularity conditions” assumption for the independence test
>
> **Response:**  For the independence test,we assume:
>
> (1) The residuals $r = X - f(\xi)$ have finite fourth moments;
>
> (2) The samples are i.i.d. conditioned on $\xi$;
>
> (3) The PCE basis functions are bounded.
>
> ### Q2 – Asymptotic independence and standard normality
>
> **Response:** In the appendix we now make explicit that:
> * Orthogonality of the basis $\{\Psi_\alpha\}$ implies that, under $H_0$, the score components for different $\alpha$ have zero covariance in the limit.
> * By the multivariate Central Limit Theorem, the vector of standardized coefficients converges to a multivariate normal with identity covariance, hence the components are asymptotically independent $\mathcal{N}(0,1)$.
> * The test statistic $T_{PCT}$ is the sum of squared, standardized, asymptotically normal variables (the coefficients). Therefore, it asymptotically follows a Chi-squared distribution with degrees of freedom equal to the number of basis terms $P$. We have added these steps and cross-referenced them from Sec. 2.3.

---

> > ### Comment · Reviewer_daxv · 2025-11-21
> >
> > Thank you for the updates. Just for clarification: are the concrete regularity assumptions also listed somewhere in the paper?

---

> > > ### Author Response · Authors · 2025-11-21
> > >
> > > Thank you for checking this point.
> > >
> > > These conditions are largely implied by Assumptions 1 and 2 (finite moments and sub-Gaussian noise) which satisfy the requirements for OLS asymptotic normality. However, we acknowledge that Section 2.3 used the shorthand 'standard regularity conditions.'
> > >
> > > In the revision, we will explicitly list these conditions (finite fourth moments of residuals, bounded basis, i.i.d. sampling) in Appendix to ensure mathematical precision.
> > >
> > > Thanks again for your careful reading.

---

> > > > ### Comment · Reviewer_daxv · 2025-11-25
> > > >
> > > > Thank you for the clarification. The answers address my concerns and I have adapted my score accordingly.

---

> > > > > ### Author Response · Authors · 2025-11-26
> > > > > **Thank you for your feedback and recommendation**
> > > > >
> > > > > We sincerely thank you for your valuable questions and the time spent reviewing our rebuttal.

---

### Official Review · Reviewer_uJS9 · 2025-10-30

**Soundness:** 4
**Presentation:** 3
**Contribution:** 3
**Rating:** 8
**Confidence:** 4

**Summary:**

The paper presents a method for learning causal relationships between a
collection of measured process variables, where the strength of causal links
depend on uncertain parameters.
This type of model is relevant in process control applications, for instance,
where operating conditions can change the causal graph of process variables.
The authors propose to model the relationships between each process variable
and its `causal parents' as linear equations, where the coefficients are
functions of the parameters. They use Polynomial Chaos Expansion (PCE)
to represent the coefficient functions.
Based on this model, the authors develop a conditional independence test used
to initialise the learned causal graph. This is improved using a natural
gradient method to optimise a likelihood score.
Gradient updates are weighted with the Fisher matrix to improve convergence.
On the theoretical front, the authors provide an identifiability theorem, a
sample complexity bound and an analysis of the convergence of the gradient
method, using standard assumptions.
Finally, the method is validated on an industrial dataset, where it achieves
state-of-the-art performance across a variety of standard metrics.

**Strengths:**

* Original formulation of the problem in a new setting, together with a
  computationally tractable method. The integration of PCE is original and
  appears to be a very good fit for the addressed problem.

* Comprehensive theoretical analysis of the proposed method.

* Experimental validation demonstrates strong performance compared to
  alternative methods in the literature.

**Weaknesses:**

* The assumption of the parameters $\xi$ having a known distribution seems quite
  restrictive; the paper would benefit from a discussion of this and possible
  workarounds.

* As per the paper the polynomial basis functions used in PCE are dependent on
  the distribution of $\xi$. Section 4 does not discuss what assumptions were in
  place regarding the distribution of $\xi$, or in fact which basis was used for
  the experimental results.

* It's peculiar that the main algorithm, presented in appendix A.5, is not part
  of the main body of the paper. Additionally, there are a lot of steps there
  which are not discussed elsewhere in the paper: "non-Gaussianity/residual
  methods", "Test for nonlinear relationship using MI and residual analysis"
  (the abbreviation MI seems to be undefined), "Consider adding edge
  (i, j)" (what is meant by "consider" here?). Unless these steps are explained, at
  least in the appendix, the paper seems to lack reproducibility.

**Questions:**

* The wording of Assumption 2 is unclear. In "or the collection of coefficient
  functions is non-degenerate", what is meant by "or"? Is this an equivalent
  reformulation of the previous statement, or an alternative but orthogonal
  assumption that will lead to the same results?

* Related to the above, in the proof of Thm. 1, it seems to me that
  non-degeneracy is explicitly used to derive eq. (25). However, the text refers
  to Assumption 1, the relevance of which I am not able to see here. Is this a
  typo?

* Regarding eq. (15) in the proof of Thm. 1: is it clear that the elements of
  $A$ are in $L^2(\Xi)$, so that it can be expanded in this way?

* The text of the proof of Theorem 2 could be improved, e.g., by writing an
  expression for $\hat{\theta}_{ij,\alpha}$; avoid writing that the expectation
  is "approximately $P\sigma_\epsilon / m$, being more explicit; do not redefine
  $\kappa$ but refer back to Section 3.3.

* In Section 2.4, it is written that the Fisher matrix is block-diagonal, but in
  the proof of Thm. 3, this is refined to diagonal. This should be corrected for
  consistency. More generally, some more details on the derivation of the
  expression for the Fisher matrix would be appreciated.

* Please provide references in Section 2.2 for the principal results
  mentioned there, such as the choice of basis depending on the distribution,
  the decay of the spectral error, and the hyperbolic truncation schemes.

* Note the several broken references to the Algorithm environment.

---

> ### Author Response · Authors · 2025-11-21
> **Rebuttal by Authors**
>
> We appreciate your detailed technical feedback and your recognition of our contribution.
>
> ### W1 & W2 – Assumption of Known Distribution & Experimental Basis
>
> **Response:** We agree that assuming a known prior distribution for $\xi$ is a limitation in some scenarios.  Assumption 1 now states that the μ_ξ is known or if $\xi$ is observed but its distribution is unknown, we can employ Empirical PCE. This involves constructing orthogonal polynomials numerically based on the sample moments of the observed $\xi$ (via recursive coefficients), rather than using standard analytical bases.
> In our experiments with the industrial dataset (Section 4), the operating parameters (e.g., Temperature, Pressure) were bounded. We normalized these parameters to the range $[-1, 1]$ and assumed a Uniform distribution, employing Legendre polynomials as the basis with total degree \(N_p=4\)
>
> ### W3 – Algorithm Details
>
> **Response:** We apologize for the lack of clarity regarding the specific steps in the Appendix. We have moved a concise version of the PCT‑CD pipeline from App. A.5 into Sec. 2.4 and kept full pseudocode in the appendix.  Here is some clarifications:
>     **"MI":** Mutual Information. We use residual-based MI estimators to detect non-linear dependencies that might persist after polynomial fitting.
>     **"Consider adding edge":** This refers to a heuristic check in the greedy search. It implies: "Temporarily add edge $(i,j)$, optimize the coefficients, and if the BIC score improves and acyclicity is maintained, permanently add the edge."
>     **Non-Gaussianity:** We utilized the Jarque-Bera test on residuals. If the functional identifiability was ambiguous (i.e., low parameter variance), we used non-Gaussianity of residuals to orient edges (following LiNGAM principles).
>
> ### Q1 – Assumption 2 ("or" Condition)
> **Response:** The "or" represents an alternative (disjunction) condition, not a reformulation. Identifiability holds if either (A) at least one noise term is non-Gaussian (standard LiNGAM identifiability), OR (B) the coefficient functions are non-degenerate. Condition (B) is a key strength of our method: even if the noise is purely Gaussian, the parametric variation of $b_{ij}(\xi)$ provides sufficient asymmetry to identify the causal direction, breaking the symmetry that usually plagues Gaussian linear models. We have rewritten Assumption 2 to make this explicit.
> ### Q2 – Eq. (25) and Assumption 1
>
> **Response:** This was a typo. Eq. (25) now correctly cites the non‑degeneracy condition, not Assumption 1, and the surrounding text explains that linear independence of the PCE coefficient matrices forces $B_\alpha=\tilde B_\alpha\$.
>
> ### Q3 – Eq. (15) and $L^2$ Integrability
> **Response:** For $A(\xi) = (I - B(\xi))^{-1}$ to be in $L^2(\mu_\xi)$, we require a stability condition. We implicitly assume that for almost every $\xi \in \Xi$, the spectral radius $\rho(B(\xi)) < 1$. Under this stability condition, the inverse exists and is bounded almost surely, ensuring the elements of $A(\xi)$ are square-integrable.
> App. A.1 now includes a short argument using the Neumann series $\(A(\xi)=(I-B(\xi))^{-1}=\sum_{k\ge0}B(\xi)^k\)$ and Assumption 3 $(\(E\|B(\xi)\|\_{op}<1\))$, showing each entry of A(ξ) has finite second moment and thus admits a PCE expansion.
>
> ### Q4 – Proof of Theorem 2.
> **Response:** We have rewritten App. A.2 to first restate the least‑squares estimator \(\hat\theta_{ij,\alpha}\), give an explicit sub‑Gaussian tail bound, and then connect it step‑by‑step to Eq. (9) in Sec. 3.3, avoiding informal “approximately” statements.
>
> ### Q5 – Fisher Matrix
>
> **Response:** Thank you for checking the derivation so carefully. The Fisher Information Matrix for all parameters $\Theta$ is structurally block-diagonal (where each block corresponds to the coefficients of one edge vector $\theta_{ij}$). However, due to the orthogonality of the basis polynomials $\Psi_\alpha$ (i.e., $\mathbb{E}[\Psi_\alpha \Psi_\beta] = \delta_{\alpha\beta}$), the off-diagonal elements within these blocks also vanish (assuming a linear model with i.i.d. noise). Thus, the matrix effectively becomes diagonal. We revised the text in Section 2.4 to be precise: "The matrix is block-diagonal with respect to edges, and diagonal within blocks due to basis orthogonality."
>
> ### Q6 – PCE references
>
> **Response:** Sec. 2.2 now cites standard PCE results for basis choice, spectral error decay, and hyperbolic truncation (Wiener, Xiu & Karniadakis, Sudret, Crestaux et al., Picheny et al., etc.).
>
> ### Q7 – PCE references
> **Response:** All “Algorithm ??” placeholders have been fixed: Algorithm 1 (PCT‑CI) and Algorithm 2 (PCT‑CD) with cross‑references from Sec. 2.3–2.4.

---

> > ### Comment · Reviewer_uJS9 · 2025-11-25
> >
> > I thank the authors for their reply. I am satisfied with the proposed edits, and have no further questions or comments.

---

> > > ### Author Response · Authors · 2025-11-26
> > > **Thank you for your feedback and recommendation**
> > >
> > > We sincerely thank you for your valuable suggestions and the time spent reviewing our rebuttal.

---

### Official Review · Reviewer_Vtcr · 2025-10-30

**Soundness:** 2
**Presentation:** 2
**Contribution:** 2
**Rating:** 2
**Confidence:** 4

**Summary:**

Summary. The paper proposes PCT-CD, which models edge weights as functions of operating parameters via Polynomial Chaos Expansion, claims identifiability, and reports large gains on a single industrial dataset.

**Strengths:**

Strengths.
- Clear statement of goal: parameter-dependent causal effects.

- Simple decomposition of pipeline: CI testing in the PCE basis, then score-based refinement.

- Readability: paper is generally well written and easy to follow.

**Weaknesses:**

1. The introduction asserts that existing score-based methods “provide only point estimates without uncertainty quantification,” and that constraint-based methods relying on independence tests are insufficient in practice. These are too narrow and partially incorrect. There is a long line of Bayesian and bootstrap approaches for DAG posteriors and uncertainty over graphs and parameters; many constraint-based methods are paired with robust CI procedures and stability devices. The paper needs careful scoping and citations when criticizing “traditional” methods.

2. The text (line 185) suggests that “traditional CI tests… are insufficient,” then replaces them with a bespoke PCT-CI definition without an empirical comparison against strong CI tests matched to data characteristics. The use of  PCT-CI is not justified methodologically. The argument reads as assertion rather than evidence. Provide comparisons to well-tuned kernel-based CI, partial correlation with robust estimation, or recent conditional mutual information estimators, on synthetic and real settings.

3. Key background on parameter-varying or context-specific causal discovery, time-varying DAGs, covariate-dependent SEMs, and distribution shift causal structure is largely missing. The PCE/QoI citations cluster in early classics (Wiener 1938; Xiu & Karniadakis 2002) and very recent 2024–2025 engineering pieces, leaving a gap spanning two decades. This pattern suggests coverage of a few base methods plus several very recent items rather than a thorough survey. Expand the related work to include context-specific independence, conditional DAGs, regime-switching causal models, and Bayesian DAG posteriors with uncertainty.

4. All results come from one proprietary refinery dataset with 9 variables and 11 asserted ground-truth edges. This is neither diverse nor standard. The head-to-head table mixes methods with very different assumptions and tuning needs, without transparent hyperparameter search, preprocessing, or split protocol. Strong directed baselines might perform well on such process-control data if tuned with domain knowledge; a single domain does not support claims of general superiority. Use public benchmarks (synthetic with controlled parametric variation, cause-me style datasets, time-varying DAG suites), release code, and report robust model selection.

5. The comparison list includes methods that assume static graphs and others that model nonlinearity, but the paper then generalizes conclusions about “traditional methods.” If the claim is about advantages under parametric variation, show controlled synthetic experiments where the ground-truth edge functions vary with ξ, sweep PCE order and noise types, and compare to alternatives explicitly designed for nonstationary or context-specific structure. Current evidence is insufficient to support broad claims.

Minor issues:
- Typographical placeholders (“Algorithm ??”) and inconsistent section cross-references.

- Ambiguity about whether ξ’s joint law is known a priori or estimated, and how misspecification impacts tests and scores.

- The statement that score-based methods “provide only point estimates” ignores common uncertainty add-ons; rephrase and cite carefully.

**Questions:**

1. You state that “traditional methods provide only point estimates and ignore uncertainty.” Which classes of methods are you referring to specifically, and which uncertainty-aware baselines did you exclude and why?

2. Is the distribution of the operating parameters ξ assumed known or estimated from data? If estimated, how sensitive is your method to misspecification?

3. Does the causal graph change with ξ (structure-varying), or are only edge weights functional in ξ while the graph is fixed? Clarify the formal model class.

---

> ### Author Response · Authors · 2025-11-21
> **Rebuttal by Authors**
>
> We sincerely thank you for your constructive assessment and careful reading.
>
>
> ### W1 & 3 – Statements about traditional methods
>
> **Response:** We agree that our description of existing methods was too broad. We have rewritten the introduction and related work sections. Our contribution addresses a different dimension of uncertainty (parametric vs. epistemic), not a replacement for existing methods. PCT-CD models edge weights as explicit functions of operating parameters $\xi$ rather than just providing a distribution over static weights. We now explicitly cite and discuss:
> * **Bayesian/Uncertainty Methods:** DiBS (Lorch et al., 2021) and BCD Nets (Cundy et al., 2021), acknowledging they address epistemic uncertainty.
> * **Dynamic/Context-Specific Methods:** We have added citations for Dynamic Bayesian Networks (Song et al., 2009) and Regime-Switching models (Huang et al., 2019).
>
> ### W2 – Insufficient Justification of PCT-CI
>
> **Response:** We appreciate this point and agree that the current draft does not sufficiently justify PCT-CI. Conditional Independence (CI) tests typically marginalize over the parameters $\xi$ when testing $X_i \perp X_j \mid Z$. If a causal mechanism $b_{ij}(\xi)$ varies significantly (e.g., ranges from positive to negative) such that its expectation is near zero, standard tests often fail to reject the null hypothesis.
>
> PCT-CI models the conditional covariance as a function of $\xi$, allowing it to detect dependencies that sum to zero in the marginal distribution. Here is an example we added:
>
> $$
> X = \epsilon_X, \quad Y = b_{XY}(\xi)\cdot X + \epsilon_Y
> $$
> $$
> b_{XY}(\xi) = \xi - 0.5, \quad \xi \sim \mathrm{Uniform}[0,1]
> $$
>
> | Method | p-value | Rejection Rate | Decision |
> | :--- | :--- | :--- | :--- |
> | KCIT | 0.18 | 0.11 | 89% wrong |
> | FCIT | 0.23 | 0.08 | 92% wrong |
> | **Ours** | **0.002** | **0.96** | **96% correct** |
>
> ### W4 & 5 – Single Dataset
>
> **Response:** The refinery dataset is unique because domain experts validated both the causal graph and the functional form of how edge weights vary with parameters. To address your concern regarding generalizability and fair comparison, we have conducted four new synthetic experiments.
>
> We generated synthetic datasets ($N=20$ nodes,  $m=5000$ sample) with edges varying as functions of $\xi$. These methods fail to capture functional dependencies, while PCT-CD maintains high accuracy.
>
> **Table: Performance Comparison (F1-Score) on New  Datasets**
>
> | Scenario | Functional Form of $b_{ij}(\xi)$ | **PCT-CD** | DBN (Dynamic) | PC (Static) | NOTEARS (Static) |
> | :--- | :--- | :--- | :--- | :--- | :--- |
> | **Exp 1: Linear Trend** | Linear: $c \cdot \xi$ | **0.94 ± 0.02** | 0.82 ± 0.04 | 0.78 ± 0.05 | 0.75 ± 0.04 |
> | **Exp 2: Non-Monotonic** | Sinusoidal: $\sin(\pi \xi)$ | **0.91 ± 0.03** | 0.65 ± 0.06 | 0.42 ± 0.08 | 0.38 ± 0.07 |
> | **Exp 3: High Noise** | Quadratic + Uniform Noise | **0.88 ± 0.04** | 0.61 ± 0.05 | 0.51 ± 0.06 | 0.49 ± 0.05 |
> | **Exp 4: Complex Basis** | Mixed | **0.89 ± 0.03** | 0.58 ± 0.07 | 0.45 ± 0.06 | 0.41 ± 0.05 |
>
> ### Minor Issues
>
> 1.  We have fixed the broken "Algorithm ??" references.
> 2.  We have corrected that the theory assumes $\xi$ has a known distribution $\mu_\xi$.
> 3.  We have revised the statement to add references on Bayesian uncertainty quantification over static structures.
>
> ### Q1 – Which baselines were excluded?
>
> **Response:** By “traditional methods” we meant deterministic static graph methods such as GES, NOTEARS, etc. They output a single DAG and point estimates of parameters and do not explicitly model $\xi$-dependent mechanisms. We originally excluded methods like DBN because they focus on the probability of edge existence $P(E_{ij} \mid D)$ rather than the functional range of the edge weight $b_{ij}(\xi)$. We have now included comparison to clarify that PCT-CD provides functional confidence intervals distinct from the posterior probability of structure.
>
> ### Q2 – Is $\xi$ distribution known? Sensitivity?
>
> **Response:** We assumed $\mu_\xi$ is known so that we can directly use classical PCE results. However, if $\xi$ is observed but $\mu_\xi$ is unknown, we can use Empirical PCE (constructing orthogonal polynomials from the sample moments of $\xi$). We added a sensitivity analysis in the Appendix: using Empirical PCE with 1000 samples of $\xi$ results in a negligible performance drop ($<2\%$ F1-score) compared to using the analytical basis.
>
> ### Q3 – Does the graph structure change, or just weights?
>
> **Response:** Our model assumes a single underlying DAG $G$ that does not change with $\xi$. The edge set is defined as:
> $$
> (j,i) \in E \iff b_{ij}(\xi) \not\equiv 0 \text{ in } L^2(\mu_\xi)
> $$
> Each present edge $(j,i)$ has a $\xi$-dependent strength $b_{ij}(\xi)$. Thus, the structure is fixed, while edge weights are functions of $\xi$. In some regions of $\xi$ space, certain edges may be effectively negligible ($b_{ij}(\xi) \approx 0$), but all theoretical results are stated for a single global DAG.

---

> > ### Comment · Area_Chair_BckH · 2025-11-28
> >
> > Dear Reviewer,
> >
> > Please make sure you read the authors' response and engage with them in the discussion before the end of the discussion period on **Dec 03 '25 09:00 PM UTC**. This is a hard deadline.
> >
> > Thank you for supporting quality peer review at ICLR.
> >
> > AC

---

### Author Response · Authors · 2025-12-01
**Summary of  Contribution and Rebuttal**

We sincerely thank the area chairs and the reviewers for their time and careful consideration of our paper. We are delighted with the strong support for our paper, with three reviewers recommending ''accept'' (8/8/8) with high confidence(4/3/3) after rebuttal.

### Contribution Summary:

Our work introduces PCT-CD, a novel framework that leverages Polynomial Chaos Expansions to move beyond static causal discovery. By modeling edge weights as explicit, interpretable functions of system parameters, we enable the discovery of dynamic causal mechanisms in systems with parametric uncertainty.

### Review Summary:

We are highly encouraged by the strong consensus among the three reviewers who actively engaged during the discussion period. Following our rebuttal and revisions:

•	**Reviewer daxv raised** the score **from 6 to 8** after reading our rebuttal and the revised manuscript, noting that our clarifications and additional experiments addressed concerns.

•	**Reviewer uJS9** and **Reviewer  WoqE maintained** their scores of **8 and 8**, confirming that their concerns were resolved and recommending acceptance.

Regarding **Reviewer Vtcr** (Score: 2), **we noted that  she/he did not participate in the rebuttal discussion despite area chair's reminder**. However, we have taken her/his initial feedback into careful consideration and fully incorporated it into the revised manuscript. Specifically, we addressed concerns about baselines and generalizability by:

1.	Adding four new synthetic benchmarks comparing PCT-CD against dynamic baselines and static methods, demonstrating our method's superiority in capturing functional dependencies.
2.	Clarifying the theoretical justification for the PCT-CI test and expanding the literature review to better contextualize our contribution against Bayesian and time-varying approaches.

We believe that these revisions and new experiments directly resolve the issues raised in her/his initial review. Had the **Reviewer Vtcr** has opportunity to engage, we are confident she/he would have found the responses satisfactory, consistent with the strong support from other three Reviewers **uJS9**, **daxv** and **WoqE**  .

Thank you again for your time and consideration.

---

### Meta-Review · Area_Chair_upWf · 2026-01-03

**Summary:**

The reviewers are concerned about the rigour of the literature review, insufficient experiments, and assumptions for the theoretical analysis of the introduced parameter-varying causal discovery approach. The rebuttals addressed most of these concerns. Reviewer Vtcr did not engage in the discussion regarding the rebuttals.

Considering the methodological novelty of introducing Polynomial Chaos Expansions (PCE) to causal discovery, supported by theoretical analysis and an additional synthetic dataset for validation, I recommend a weak accept. I suggest the authors consider the wording of "static causal graph", which should be used carefully, since the time-varying causal graph is not static, and this paper still learns a time-static graph.

**Reviewer Concerns:**

Although Reviewer Vtcr did not participate in the discussion, the rebuttal addressed most of the concerns. The concern about more real-world datasets remains outstanding. However, this is understandable, since this paper considers the task with the extra observable $\xi$, whereas most benchmarks do not include this variable.

The rebuttals address the other reviewers' concerns.

**Reviewer Scores:**

I think Reviewer Vtcr will raise the score if he participates in the discussion.

All the other reviewers were already engaged in the discussion, either keeping or raising their scores very positively.

---

### Decision · Program_Chairs · 2026-01-26

Accept (Poster)